# AIRDELHI: Fine-Grained Spatio-Temporal Particulate Matter Dataset From Delhi For ML based Modeling

**Sachin Kumar Chauhan, Sayan Ranu, Rijurekha Sen**
Department of Computer Science
IIT Delhi
{csz188012, sayanranu, riju}@cse.iitd.ac.in

**Zeel B Patel, Nipun Batra**
Department of Computer Science
IIT Gandhinagar
{patel_zeel, nipun.batra}@iitgn.ac.in

## Abstract

Air pollution poses serious health concerns in developing countries, such as India, necessitating large-scale measurement for correlation analysis, policy recommendations, and informed decision-making. However, fine-grained data collection is costly. Specifically, static sensors for pollution measurement cost several thousand dollars per unit, leading to inadequate deployment and coverage. To complement the existing sparse static sensor network, we propose a mobile sensor network utilizing lower-cost $PM_{2.5}$ sensors mounted on public buses in the Delhi-NCR region of India. Through this exercise, we introduce a novel dataset AIRDELHI comprising $PM_{2.5}$ and $PM_{10}$ measurements. This dataset is made publicly available at *https://www.cse.iitd.ac.in/pollutiondata*, serving as a valuable resource for machine learning (ML) researchers and environmentalists. We present three key contributions with the release of this dataset. Firstly, through in-depth statistical analysis, we demonstrate that the released dataset significantly differs from existing pollution datasets, highlighting its uniqueness and potential for new insights. Secondly, the dataset quality been validated against existing expensive sensors. Thirdly, we conduct a benchmarking exercise (*https://github.com/sachin-iitd/DelhiPMDatasetBenchmark*), evaluating state-of-the-art methods for interpolation, feature imputation, and forecasting on this dataset, which is the largest publicly available PM dataset to date. The results of the benchmarking exercise underscore the substantial disparities in accuracy between the proposed dataset and other publicly available datasets. This finding highlights the complexity and richness of our dataset, emphasizing its value for advancing research in the field of air pollution.

## 1 Introduction

United Nations Sustainable Development Goals [UN, 2015], especially the Goal-11 Sustainable Cities and Communities, is a primary research focus in institutions of developing countries like India [IIITD, 2023], with related research works in sustainable transport [Chauhan *et al.*, 2020] and pollution [Shukla *et al.*, 2020; Bikkina *et al.*, 2019; Iyer *et al.*, 2022; Abidi *et al.*, 2022].

Air pollution has now reached life-threatening levels in Delhi-National Capital Region (NCR), India [Tripathi *et al.*, 2019; Mannucci and Franchini, 2017], which is one of the most densely populated urban centers. The population of Delhi-NCR exceeds 46 million people [Nagar *et al.*, 2017] and it has been reported that 50% of all children staying in this region suffer from irreversible lung damage [Chatterji, 2021; ORF, 2021]. *Particulate Matter (PM)* is especially dangerous, since

our breathing cannot filter out the ultra-fine particles. To mitigate the effects of air pollution, there is an urgent need to identify causes of pollution and strategies to curb its spread. [Sahu *et al.*, 2020; Sutaria, 2022] suggests to use one sensor per km$^2$ for better pollution analysis. The *Central Pollution Control Board (CPCB)* and *Delhi Pollution Control Committee (DPCC)* have only 81 realtime air pollution measurement centers in Delhi-NCR Sutaria [2022] with 65 manually monitored centers, which are thoroughly inadequate [Guttikunda *et al.*, 2023; ET, 2022] to cover the vast geography of $55,000$ km$^2$ [NCRPB, 2018].

In the literature, several models have been proposed for predicting pollution levels at same/future time points [Patel *et al.*, 2022; Gao and Li, 2021; Kurt *et al.*, 2008; Tsai *et al.*, 2018; Le *et al.*, 2020], and identifying factors affecting pollution [Apte *et al.*, 2011; Google, 2014; Messier *et al.*, 2018; Apte *et al.*, 2017; Alexeeff *et al.*, 2018]. There exists *interpolation models* [Qiao *et al.*, 2019; Ras and Williams, 2005; Hamilton *et al.*, 2017; Patel *et al.*, 2022] to reliably predict pollution levels at unseen locations based on a sufficient number of pre-installed sensors. These models can improve with fine-grained pollution data. The interpolation and forecasting models are *supervised* in nature and hence can do better with more training data. Unfortunately, collecting pollution data using realtime centers is highly expensive as each instrument costs thousands of US Dollars.

In this work, we aim to mitigate the problem of lack of sufficient data in a cost-effective manner. We design a low-cost sensing mechanism (thoroughly compared in quality against high cost sensors) that allows us to collect PM data over a subset of the Delhi-NCR region at a fine spatio-temporal granularity. The key highlights and contributions of our work are:

**1. Quality dataset:** As it is not cost-effective to repeat even the low cost sensors per km$^2$, we establish a low-cost vehicle-mounted PM sensing network and release the largest PM$_{2.5}$ dataset from one of the most polluted regions in the world. This dataset is shown to be as good as the data collected from the few high-cost static-sensor deployed in the same region. As it is very challenging to collect such dataset in a developing country due to constraints in infrastructure and government permissions, we document our data collection experience briefly in the paper. (§ 3.2).

**2. Unique dataset:** This dataset complements the static sensor data available from the government deployed instruments in important ways. The static sensors are located at the top of high towers to get precise recordings of ambient pollution values, not affected by local sources. Our mobile sensors, on the other hand, are installed in the bus driver's cabin to measures the ground level pollution that daily commuters breathe in. We also perform a thorough comparison with PM datasets available from other parts of the world and establish that the released dataset is unique in terms of scale and statistical characteristics. Hence, it can be of immense value to environmental think tanks. (§ 3.3).

**3. Utility for ML modeling:** Through extensive benchmarking using state-of-the-art Machine Learning (ML) algorithms, we demonstrate the utility of this new dataset for modeling problems using ML, like spatio-temporal interpolation, missing data imputation and forecasting. The dataset is shown to be more challenging to model with ML algorithms, compared to previously available datasets, as Delhi has much higher variance in PM across space and time. This dataset, therefore opens opportunities for ML researchers for designing and benchmarking new ML algorithms, to reduce the interpolation, missing data imputation or forecasting errors. (§ 4).

## 2 Related Work

A primer for Air Pollution Monitoring is available at Urbanemissions [2023]. Spatio-temporal (ST) interpolation involves predicting air quality at unmonitored locations in the past and/or present time using training data observed from the sensors during the past and present time. Zheng *et al.* [2013] developed a co-training-based approach for ST interpolation using PM$_{2.5}$ values captured every hour from ground stations of 4 cities in China which are converted to AQI (Air Quality Index), along with meteorological and traffic data. Cheng *et al.* [2018] proposed an attention-based hybrid model involving LSTM and dense layers and Patel *et al.* [2022] proposed a domain-inspired non-stationary Gaussian process model for ST interpolation which can also be used for ST forecasting. The two used 36 monitoring stations in Beijing with the collection time interval of 1 hour (with the latter additionally using London data), alongside meteorological data.

Missing data imputation problem can be considered a variation of spatio-temporal interpolation where observations on the spatio-temporal cube are missing at random and we want to impute the missing data. Models that work for ST interpolation can mostly be adapted readily for this problem.

Spatio-temporal forecasting aims to predict air quality at a particular location in future using the past and current data available at all the installed sensors. Kurt *et al.* [2008] developed an online neural network based approach to predict air quality maximum 3 days ahead in time using 1 year PM$_{10}$ data

for 1 region in Turkey. Zheng *et al.* [2015] develop and deploy a machine learning based air quality forecasting system with the Chinese Ministry of Environmental Protection. Yi *et al.* [2018] develop a deep learning based approach to provide short-term and long-term air quality forecasts. The two used meteorological data along with pollution data generated every hour from 2,296 stations in 302 Chinese cities, and converted these concentrations into corresponding (individual) AQIs according to Chinese AQI standards. Air quality forecasting was posed as a challenge in KDD2018, where Luo *et al.* [2019] presented a winning solution based on a combination of classical machine learning and deep learning models using the provided data from stations in Beijing and London. Gao and Li [2021] propose a graph-based LSTM model for air quality forecasting and evaluate on Northwest China hourly data from 32 china stations. All these prior arts utilize the static ground stations Air Quality data for the analysis, which enforces a restricted spatial coverage. They also use meteorological data from the respective regions. Bhattacharyya *et al.* [2022] provides a similar USA AQI dataset collected from Air Quality Open Data Platform.

There also have been studies on low cost sensors available in market for developed (EU) regions Karagulian *et al.* [2019]. Also, projects about installing low cost sensors at different roadside locations, like Schneider *et al.* [2023] and Iyer *et al.* [2022], to complement the existing expensive static sensor network are done recently, but they kept the sensors at fixed locations. Vehicle mounted air pollution sensing has also been conducted [Apte *et al.*, 2011; Google, 2014; Apte *et al.*, 2017; Alexeeff *et al.*, 2018; Guo *et al.*, 2016; Adams and Corr, 2019; Li *et al.*, 2012], with certain limitations. Abidi *et al.* [2022] used similar low-cost sensors to analyze the relation of PM with static (green cover, buildup, commercial, residential) and dynamic (meteorological, traffic) factors, particularly at traffic intersections with odd-even policy in Delhi. As per Yi *et al.* [2015]; Pavani and Rao [2017], static sensors are better in data quality, endurance and temporal resolution, a low-cost static sensor is better in temporal resolution which can provide (static) pollution maps with better temporal resolution for the installed locations. But the effective static pollution maps require careful placement of sensor nodes in large quantity leading to resource wastage. Mobile sensors are better in mobility, geographic coverage, maintenance, cost-efficiency which can provide (dynamic) pollution maps with greater geographic coverage, with limitation of communication overheads and redundant sampling. In contrast, we are working with the PM data, collected with mobile sensors deployed on frequent bus routes, which is fine-grained and provides better spatio-temporal coverage, and our benchmarked models do not rely on meteorological factors.

## 3 Dataset Description

### 3.1 Dataset Collection Challenges

Creating the mobile PM dataset (as a replacement for low cost static PM dataset and high-cost ground station PM dataset) required us to design and implement our own embedded platform, choosing and calibrating appropriate sensors for maximum accuracy at low cost. The complete design is presented in Goyal *et al.* [2018]. We opted to install our device in public buses, to utilize their pre-defined/fixed and frequent routes of travel. Packaging was challenging to securely mount the instruments in the public buses, avoiding theft and ensuring enough ambient air to measure PM. Cellular connectivity was intermittent as the buses traversed the city, requiring us to augment real time data transfer when signal was present, with local storage to save data when signal strength dropped. Finally, getting permissions from different government entities to instrument the public bus fleet needed strict safety certifications that our devices do not interfere with the electrical and mechanical functioning of bus.

We mounted pollution tracking sensors on the permissible 13 public buses in Delhi for 3 months (Nov 1, 2020 to Jan 31, 2021), in collaboration with Delhi Integrated Multimodal Transport System, after rigorous tests for automotive safety certification and appropriate permissions and letters of support from the Delhi Ministry of Transport and Delhi Pollution Control Committee (provided in Appendix I). The Covid'19 restrictions were relaxed by this time, limiting to containment zones at local level only [MHA, 2020]. So, the impact of the COVID'19 on the collected dataset is expected to be minimal, if not none.

As discussed in Goyal *et al.* [2018], the inside of our custom-made instrument comprising *(a)* PM sensor measuring $PM_{2.5}$, $PM_{10}$ and $PM_1$, *(b)* GPS sensor to locate the bus, *(c)* 4G radio to communicate data from bus to server, *(d)* SD card for locally storing data when 4G signal is unavailable, *(e)* BME sensor [BME, 2023], a sensor especially developed for mobile applications and wearables, to record temperature and relative humidity and *(f)* micro-controller to orchestrate the sense-store-communicate software (See Fig. 1a). The mounting location in the bus driver's cabin, next to two open windows to allow enough air-flow (Fig. 1b-1c). Each bus commutes for

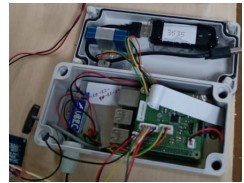 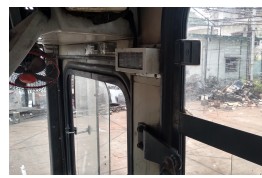 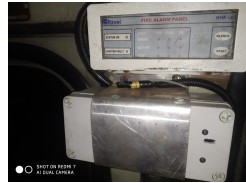 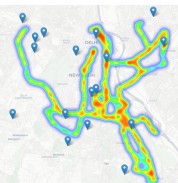

(a) Measuring device     (b) Mounting location     (c) Mounted device     (d) Bus trajectories

Figure 1: (a) Inside of our PM measuring IoT unit. (b) Mounting location in bus driver's cabin in non air-conditioned public bus (below the existing white box). (c) Mounted IoT unit in the bus (below the existing white box). (d) Government deployed static sensors installed in and around our bus trajectories, as location icons.

16-20 hours per day, and our instruments collect data at a fine granularity of 20 samples per minute. Overall, the bus trajectories cover 559 km$^2$, along the main arterial roads in North-West, North, North-East and South-East Delhi (Fig. 1d). The dataset, having 3 pollution parameters: $PM_1$, $PM_{2.5}$ and $PM_{10}$ with 7 non-pollution parameters: latitude, longitude, time, deviceId, pressure, temperature, relative humidity, has been made available at `https://www.cse.iitd.ac.in/pollutiondata/` and `https://huggingface.co/sachin-iitd/DelhiPollDataset` with proper documentation, under a Creative Commons Attribution 4.0 International License [CC-by4, 2013].

## 3.2 Data Quality Analysis

Fig. 2a plots $PM_{2.5}$ values measured by two low cost PM sensors built by us (cost USD 30), and the same measured by an industry grade reference instrument TSI DustTrak (cost USD 9500), while all three instruments are placed close to each other. The plot shows hours of the day along $x$-axis and sensed $PM_{2.5}$ values along $y$-axis, for 10 sample days Jul 21-31, 2021. This is after the deployment of the low cost sensors in the buses is over, and the sensors have been brought back to the lab.

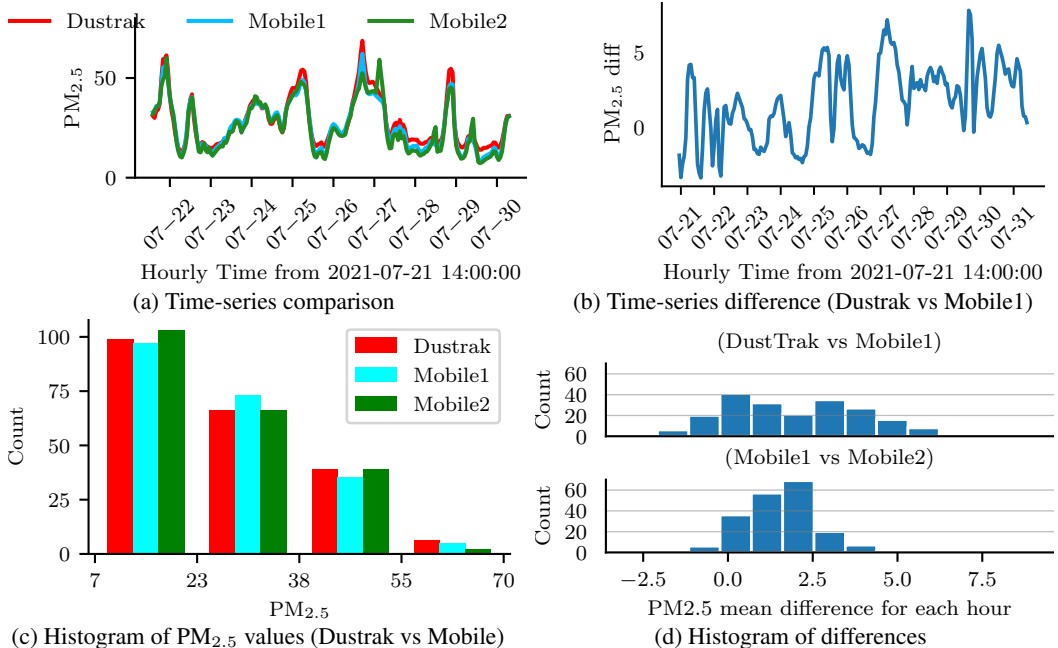

(a) Time-series comparison

(b) Time-series difference (Dustrak vs Mobile1)

(c) Histogram of $PM_{2.5}$ values (Dustrak vs Mobile)

(d) Histogram of differences

Figure 2: (a) $PM_{2.5}$ values measured by our low-cost mobile PM sensors (USD 30) vs. TSI DustTrak (USD 9500) between Jul 21-31, 2021. (b) Mean Difference and (d) Histogram of pointwise differences of $PM_{2.5}$ values measured by DustTrak and low cost mobile PM sensors. (c) Histogram of $PM_{2.5}$ in the intervals on x-axis. The values are almost identical.

Fig. 2b shows the difference of hourly mean $PM_{2.5}$ between DustTrak and one mobile sensor, the mean difference is 6.16%. Also, Fig. 2c shows the histograms of hourly mean $PM_{2.5}$ for the shown $PM_{2.5}$ intervals for the three devices. Fig. 2d shows the histogram of difference of hourly mean $PM_{2.5}$ between DustTrak and one mobile sensor, and two mobile sensors, for the same 10 days. Also, Fig. 9 in Appendix C shows the similar mean and standard deviation between the 3 devices. While the cost gap between the instruments is huge, the gap between their sensed $PM_{2.5}$ values, as seen in the plots, is negligible. This pattern has been observed consistently by us and other researchers [Zheng *et al.*, 2018; Cheng *et al.*, 2014; Gao *et al.*, 2015; Rai *et al.*, 2017; Jiao *et al.*, 2016; Zheng *et al.*, 2019].

We additionally compare the distribution of $PM_{2.5}$ values recorded by our mobile sensors vs. those by the high-cost static sensors, deployed at sparse locations by CPCB and DPCC in Delhi-NCR. Fig. 3a(Left) shows hours of day along x-axis and average $PM_{2.5}$ for that hour, as measured by reference grade static monitors, with standard-deviation bars along y-axis. Fig. 3a(Right) shows the same averaged over all bus mounted sensors. We select the static sensors that are within 1km of mobile sensor trajectory for each hour, and plot for 7 sample days. Fig. 3a reveal that both static and bus mounted sensors show similar PM distributions for each day, in spite of the difference in heights they have been installed at, and the difference in PM measurement technique. We see this agreement for the entire 3 months deployment period. The agreement between low cost mobile sensors, and a co-located high cost TSI Dusttrak, as well as reference grade static monitors, give us confidence to release the dataset to the research community.

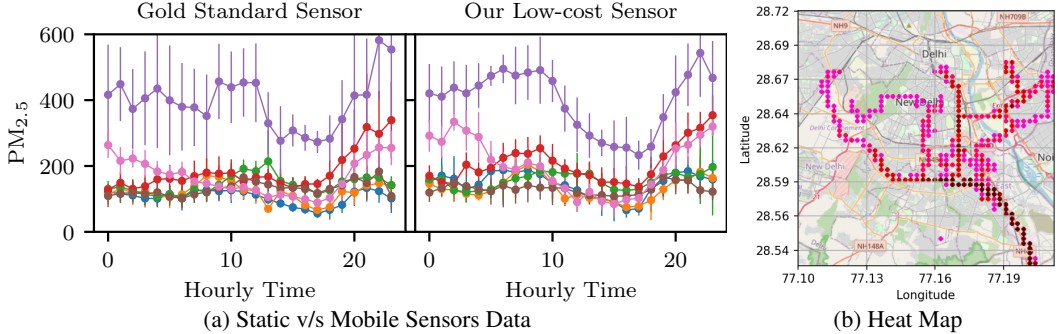

(a) Static v/s Mobile Sensors Data        (b) Heat Map

Figure 3: (a) Distribution of $PM_{2.5}$ collected by our low-cost sensor and gold standard sensor over 7 random days. The distributions are similar across the two sets of instruments. (b) Heat Map for all bus routes for Dec 15, 2020 (darker locations contain more samples).

**Heat Map:** During our analysis, we discovered variations in data availability across different timestamps and spatial locations. It was evident that certain timestamps were not available at all spatial locations. Furthermore, some spatial locations, which were situated along routes with fewer bus visits throughout the day, exhibited limited temporal samples. As illustrated in Fig. 3b, a typical day (Dec 15, 2020) demonstrated this pattern, where the outermost locations (depicted in light/pink color) contained samples from 4 hours duration within the 16.5-hour effective temporal window. Conversely, the darker/brown locations near the bottom right of the figure displayed a higher number of samples, ranging from 14 to 16.5 hours. These locations are associated with common bus routes that connect with the depot.

### 3.3 Dataset Novelty

Tables 1 and 2 summarize the statistics of the dataset. While vehicle mounted air pollution sensing has been conducted [Apte *et al.*, 2011; Google, 2014; Apte *et al.*, 2017; Alexeeff *et al.*, 2018; Guo *et al.*, 2016; Adams and Corr, 2019; Li *et al.*, 2012], our dataset is unique in characteristics and scale. Specifically, only two studies from Ontario, Canada [Adams and Corr, 2019] and Zurich, Switzerland [Li *et al.*, 2012] have made their datasets publicly available. The Zurich dataset does not include PM values. Compared to the Canada dataset, our dataset is 1000 times larger and has a significantly different distribution of PM values (See Tables 1 and 2).

This is understandable as Delhi-NCR is an air pollution hotspot, whereas Zurich and Ontario have negligible PM levels. We also compare our dataset with a recent USA AQI dataset Bhattacharyya *et al.* [2022] collected from Air Quality Open Data Platform. We also purchased Wind Speed (WS) data

Table 1: Details of Delhi, India and Hamilton, Ontario, Canada and USA datasets.

| Metric | Delhi-NCR | Canada | USA |
|---|---|---|---|
| **Total area** | 559 km$^2$ | 1138 km$^2$ | 54 cities |
| **Total samples** | 12,542,183 | 46,080 | 35,596 |
| **Samples with PM2.5** | 12,542,183 | 12,154 | 35,134 |
| **Pollutants covered** | $PM_1$, $PM_{2.5}$ and $PM_{10}$ | CO, NO, $NO_2$, $SO_2$, $O_3$, $PM_1$, $PM_{2.5}$ and $PM_{10}$ | CO, $NO_2$, $SO_2$, $O_3$, $PM_{2.5}$ and $PM_{10}$ |
| **Meteorological** | Temp, RH, Pressure, Wind Speed * | - | Temp, RH, Pressure, Dew, Wind Speed, Wind Gust |
| **Sensor source** | Public bus | Commercial van | OpenDataPlatform |
| **Monitoring days** | 91 | 114 | 668 |

Table 2: Statistical comparison of PM values in Delhi, Canada and USA datasets.

| Metric | Delhi-NCR | | | Canada | | | USA | |
|---|---|---|---|---|---|---|---|---|
| | $PM_1$ | $PM_{2.5}$ | $PM_{10}$ | $PM_1$ | $PM_{2.5}$ | $PM_{10}$ | $PM_{2.5}$ AQI | $PM_{10}$ AQI |
| **Mean** | 120.35 | 207.92 | 226.11 | 12.15 | 15.08 | 46.45 | 31.15 | 17.67 |
| **Std-dev** | 57.27 | 114.36 | 123.86 | 9.02 | 12.87 | 97.36 | 17.11 | 11.00 |
| **Missing** % | 0 | 0 | 0 | 71.71 | 73.62 | 72.24 | 1.30 | 52.34 |

for Nov 2020 - Jan 2021 from `www.windfinder.com` to complement our Delhi-NCR dataset for the meteorological analysis performed in Appendix E and Fig. 15, showing pollutants correlation with different meteorological factors and peculiar situations due to external factors (like impact of stubble burning episodes). Due to this behaviour, interpolation and forecasting are hard problems. Still, in Appendix D and Fig. 14, we observed covariance among some spatial locations, and in Fig. 11, we observed some autocorrelation over the entire dataset. Similar trends were observed with the bus route analysis shown in Appendix F. Hence we decided to formulate an interpolation problem and a 24 hour forecasting problem to analyze the model performance, next.

# 4 ML Modeling Benchmarks

In this section, we benchmark the machine learning problems of **(1)** spatio-temporal interpolation, **(2)** spatio-temporal data imputation and **(3)** spatio-temporal forecasting on the proposed, Canada and USA datasets. This benchmarking study serves two roles. First, it allows us to compare the complexities of the three datasets beyond just statistical characterization. Secondly, spatio-temporal interpolations, data imputations, and forecasting methods are crucial for environmental research, policy-making, and individual decision-making. They empower various stakeholders to gain a comprehensive understanding of air pollution, proactively address potential increases in pollution levels, and make informed choices to reduce personal exposure. In order to harness the full potential of spatio-temporal forecasting, interpolations, and data imputations, it is crucial to benchmark and evaluate the performance of algorithms designed to tackle these problems.

## 4.1 Formulation of different ML Prediction Problems

**(a) Spatio-temporal Interpolation:** Given set of visible/available locations with input features (latitude, longitude and time) and $PM_{2.5}$ available for T+1 days, we wish to estimate $PM_{2.5}$ for a set of held-out locations for the T+$1^{th}$ day using the input features (latitude, longitude and time). This approach is compatible to the scenario where we have data for some locations and we use interpolation algorithms to know the $PM_{2.5}$ values at new locations.

**(b) Spatio-temporal Missing Data Imputation:** Given set of locations with input features (latitude, longitude and time) and $PM_{2.5}$ available for T days and a set of visible locations with input features (latitude, longitude and time) and $PM_{2.5}$ available for the T+$1^{th}$ day, we wish to estimate $PM_{2.5}$ for a set of held-out locations for the T+$1^{th}$ day using the input features (latitude, longitude and time). This setting is compatible to the scenario where we have intermittent data missing throughout the day and we use interpolation algorithms to predict the missing points taking past & present data as input.

**(c) Spatio-temporal Forecasting:** Given a set of all locations with input features (latitude, longitude and time) and $PM_{2.5}$ available for T days, we wish to estimate $PM_{2.5}$ for a set of all locations for the T+$1^{th}$ day using the input features (latitude, longitude and time).

## 4.2 ML Algorithms Benchmarked in this Paper

**(a) Mean Predictor** is the simple mean value of all visible samples which is used as the value of all the held-out locations. The mean value of all visible $PM_{2.5}$ locations C is used as the value of the held-out $PM_{2.5}$ locations P.

$PM_{2.5}^p \leftarrow mean \ \forall p \in P$, where $mean \leftarrow \frac{1}{|C|} \sum PM_{2.5}^c \ \forall c \in C$

**(b) Inverse Distance Weighting (IDW)** is the weighted average value of all visible C samples in terms of distance, which is used as the value of the held-out P locations.

$PM_{2.5}^p \leftarrow \sum \frac{PM_{2.5}^c}{F(d_{cp})} \ \forall c \in C \ \forall p \in P$, where F is a linear function on distance d.

**(c) Random Forest (RF)** is a non-linear model capable of modeling complex spaces. It is known to perform efficiently on non-linear regression tasks, using an ensemble of multiple decision trees, taking the final output as the mean of the output from all trees.

**(d) XGBoost (XGB)** iteratively combines the results from weak estimators. It uses gradient descent while adding new trees during training.

**(e) ARIMA** or Auto-Regressive Integrated Moving Average is a statistical time-series forecasting model that uses linear regression. It is configured using parameters $(p, d, q)$ as: $p$ is the number of lag observations included in the model, $d$ is the number of times raw observations are differenced, and $q$ is the size of the moving average window. We use ARIMA with parameters (3, 1, 1).

**(f) N-BEATS** is Neural Basis Expansion Analysis for Time Series, a deep learning model for zero-shot time-series forecasting [Oreshkin *et al.*, 2020]. We use the code from Python library "Darts".

**(g) Non-Stationary Gaussian Process (NSGP)** is a recent gaussian processes baseline [Patel *et al.*, 2022]. It learns a non-stationary covariance [Plagemann *et al.*, 2008] for latitude and longitude and locally periodic covariance for time. In general, Gaussian process Ras and Williams [2005] a.k.a. Kriging is a Bayesian non-parametric model known as the best unbiased predictor in spatial interpolation domain. With only three tunable parameters, it is considered a strong baseline.

**(h) Graphsage** is a graph neural network model to learn and predict values at unknown spatio-temporal locations [Hamilton *et al.*, 2017]. We transform the PM data to a graph, and use Graphsage for interpolation and missing data imputation. Our graph formulation is available in Appendix A.

### 4.3 Dataset Pre-processing and Evaluation Metrics for the Analysis

To benchmark interpolation and missing data interpolation ML modeling algorithms, we split the data into two parts for *visible* and *held-out/hidden* to use the held-out part for testing purpose. For forecasting based ML modeling, the visible/held-out split is not required, as predictions are made for future timestamps for all locations. For the Delhi dataset, we focus on the data collected from Nov 12, 2020, to Jan 30, 2021, excluding the initial days when there were fewer instruments on the buses and limited sample data. Additionally, we exclude the nightly data between 10 PM IST and 5:30 AM IST when buses remain stationary at confined bus-depots. To facilitate analysis, we divide the geographical area into square spatial grids with a side length of 1 km. These grids are further converted into spatio-temporal cells with a time interval of 30 minutes. To obtain representative PM values, we compute the average of all samples within each spatio-temporal cell. Subsequently, we employ $K$-fold cross-validation to partition the data into $K$ $PM_{2.5}$ sets for each day, out of which 1 set is reserved for test and others used in training. The results obtained from the Delhi dataset are denoted as *Delhi (Day)* in the generated plots.

Additionally, we utilize two open-sources $PM_{2.5}$ datasets, from Hamilton in Ontario, Canada [Adams and Corr, 2019] and from USA [Bhattacharyya *et al.*, 2022]. For the Canada dataset, we process the data from 18 distinct days in the year 2015 using the same methodology. These results are presented as *Canada (Day)* in the respective experiments. As the data for Canada exhibits temporal sparsity, we project the data for each year onto a single day and treat it as equivalent to 11 days (from 2006 to 2016). The outcomes of this processing approach are depicted as *Canada (Year)* in the experiments. For the USA data, we use the available $PM_{2.5}$ data across 54 cities from Jan 1, 2019 to Dec 11, 2020, and the results are presented as *USA (Day)*. We benchmark the datasets on Nvidia DGX Workstation (with 4X Tesla V100 GPUs) and the benchmarking code is available at `https://github.com/sachin-iitd/DelhiPMDatasetBenchmark`.

**Evaluation Metrics:** The Loss is computed as:
$$RMSE(L'_p, L_p) = \sqrt{\frac{1}{N}\sum_{i=1}^{N}(y'_i - y_i)^2} \tag{1}$$

where $y'_i$ is the predicted and $y_i$ is the true $PM_{2.5}$ value, and $N$ is the total number of samples.

For each of the $K$ folds, we separately compute RMSE for that fold, and then plot average with standard deviation bars over the $K$ folds. The lower RMSE being the better.

**Notation:** We use T (consecutive) days data for the training and take the next day for test/evaluation. For interpolation benchmarking, Fig. 4a denotes the various subsets of this T+1 days data as A, B, C and P, for one fold of $K = 5$ fold validation. The P is the set over which *Prediction* is to be done. the C set can be understood as the *Context* for prediction. For a given fold, A is the visible set with 80% of all T train days data, B is the held-out set with the remaining 20% of the T train days data. $A \cup B$ forms the whole dataset for the T train days. Similarly, C is the visible set with 80% of the test day data, P is the held-out set with the remaining 20% of the test day and $C \cup P$ forms the whole dataset for the test day. The held-out locations B and P will be different for each of the $K$ folds. The

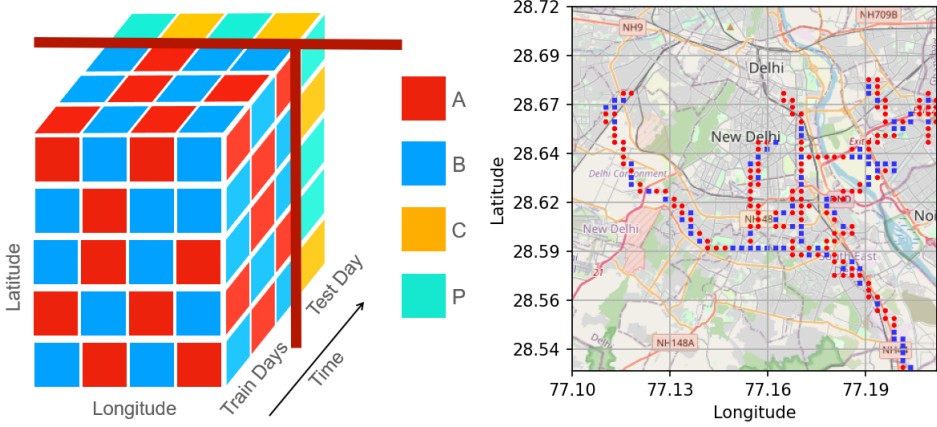

(a) An arbitrary split of A, B, C and P sets.  (b) Visible (A) and Held-out (B) sets over map.

Figure 4: PM Data Splits. (a) $A \cup B$ forms the whole dataset for the T train days, $C \cup P$ forms the whole dataset for the test day. Spatial locations of C or P are independent of spatial locations of A or B. (b) Set of A and B spatial locations for 3 PM to 4:30 PM on Dec 15, 2020 from all bus-routes.

A and C though both being visible sets, are independent of each other. The exact number of locations in A, B, C and P change across the $K$ folds. In Fig. 4b, we show set of A and B spatial locations in Delhi dataset for 3 PM to 4:30 PM on Dec 15, 2020.

## 4.4 Observations and Inferences

Fig. 5, shows the RMSE for interpolation, using 5-fold cross validation for the two training configurations ACT in Fig. 5a and C in Fig. 5b, for 3 training days. ACT uses the visible set from both training and test days, while C uses only the test day's $PM_{2.5}$ visible set. The missing data imputation plots are almost identical to the interpolation plots, so we omit these for space constraints.

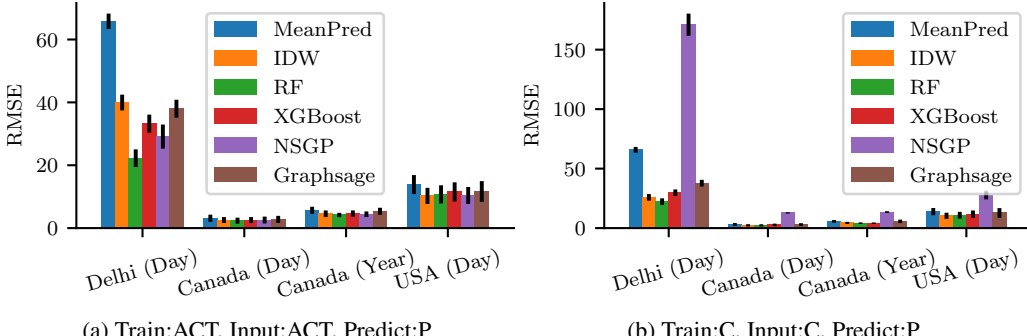

(a) Train:ACT, Input:ACT, Predict:P  (b) Train:C, Input:C, Predict:P

Figure 5: Interpolation RMSE. Training days' data is used by ML model in (a) and not used in (b).

**Observation 1: Delhi dataset is harder to model.** All experiments over Delhi data show higher RMSE and all experiments over Canada and USA data show low RMSE, for both interpolation and forecasting, in Figures 5, and 7. This shows that Delhi data is more challenging for ML modeling, than the currently available PM datasets. Also, the majority of samples present in the data is below 800 $PM_{2.5}$. Omitting the 29 test samples with higher $PM_{2.5}$ for the purview of low-cost sensor accuracy, in Fig. 6 we can see the MAE and RMSE for different $PM_{2.5}$ levels for Random Forest (RF) algorithm. We observe that the modeling errors increase with increasing $PM_{2.5}$ levels. For existing dataset like Canada with a very low $PM_{2.5}$ of mean $15 \pm 13$ (refer the Table 2), the expected modeling error would automatically be low, making the data easier to model. For the Delhi dataset with high $PM_{2.5}$ of mean $208 \pm 114$, makes the dataset hard to model.

**Observation 2: Learning from data helps in modeling the Delhi dataset.** All ML based algorithms show significant improvement over Mean Predictor for Delhi data in Figures 5a, whereas improvement for Canada and USA data over Mean Predictor is not significant. In Fig. 5a, all ML algorithms exhibit less than 40 RMSE while Mean Predictor RMSE is 65.80 for Delhi data (best case improvement is 66.2% for RF and worst case 39.3% for IDW). For Canada data, best case improvement is $\sim 27\%$ and worst case sees no improvement, whereas for USA AQI data, improvement is within 16% - 26%.

**Observation 3: Traditional ML algorithms do as well as the recent models for the Delhi dataset.** Learning from data matters, as the ML based models do better than the mean predictor. But the recent

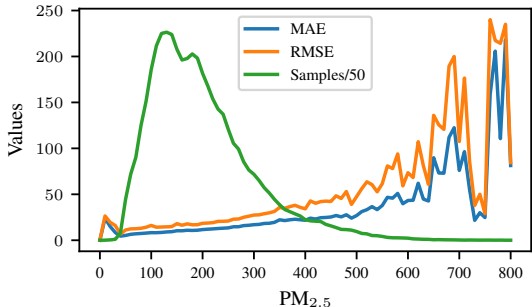

Figure 6: Errors for different $PM_{2.5}$ levels (at intervals of 10) for RF algorithm, with the number of test samples. Modeling shows high errors at high $PM_{2.5}$, making Delhi dataset hard to model due to high $PM_{2.5}$ levels.

complex Bayesian models like NSGP, and the neural network based models like GraphSage (for interpolation) and N-BEATS (for forecasting), do not outperform powerful traditional ML models like Random Forest. For instance, RF performs best for interpolation (RMSE 22.24 in Fig. 5), and XGBoost performs best for forecasting (RMSE 84.15 in Fig. 7).

**Observation 4: Historical training data adds no value for interpolation.** For the spatio-temporal interpolation problem, just using data from the visible set C from test day is enough to predict the held-out P data with low RMSE. For example, the RMSE for RF is similar (22.24) for test day only data C in Fig. 5b and with including train day data ACT in Fig. 5a. And XGBoost is better for C with RMSE 29.73 than for ACT with RMSE 33.24. NSGP is the only algorithm, which sees a huge jump in RMSE when not using training data from past days. Thus PM for a given day is mostly unrelated to PM on past days, and using historical training data has no significant impact on interpolation RMSE.

**Observation 5: A location's air quality is related to nearby location.** In Fig. 5, for Delhi data, we can observe that IDW performs significantly better than the Mean Predictor. Mean Predictor does not take the distance into account, whereas the IDW gives more weightage to nearby locations data. So, the impact of an adjacent location is significant for IDW w.r.t Mean Predictor, pointing that nearby locations air quality impacts the air quality of a location.

Fig. 7 shows RMSE of forecasting. Graphsage does not work in this setting as it requires a subset of test day's data for edge formation to the data being predicted. So we drop Graphsage, and add two forecasting specific baselines: ARIMA and N-BEATS, that are not suitable for interpolation.

**Observation 6: Forecasting is a harder problem than interpolation.** Forecasting RMSEs are significantly higher than interpolation RMSEs. The best model in forecasting is XGBoost in Fig. 7 with RMSE 84.15, whereas the best model for interpolation in Fig. 5 is RF with RMSE 22.24. Higher forecasting RMSE compared to interpolation also supports that previous day's data has less impact on test day's PM data. Hence forecasting using only past days' data for an unseen future test day is hard.

**Observation 7: How time is normalized affects forecasting accuracy.** In Fig. 7a, time normalization is done across days, i.e. time starts at 0 on first train day and increases to 1 till last train day. ARIMA / N-BEATS don't normalize the time directly, they take all PM values in a sequence corresponding to time from start to end. RF/XGBoost takes input in random sequence and hence takes the time as a state parameter, which can be normalized from start to end, or for each day. Table 7b compares this time normalization across days (T), to normalizing separately for each day. RF, XGBoost and NSGP show lower RMSE for separate normalization for each day, while IDW does better with normalization across days. This pre-processing step of time normalization therefore should be carefully decided based on the ML algorithm.

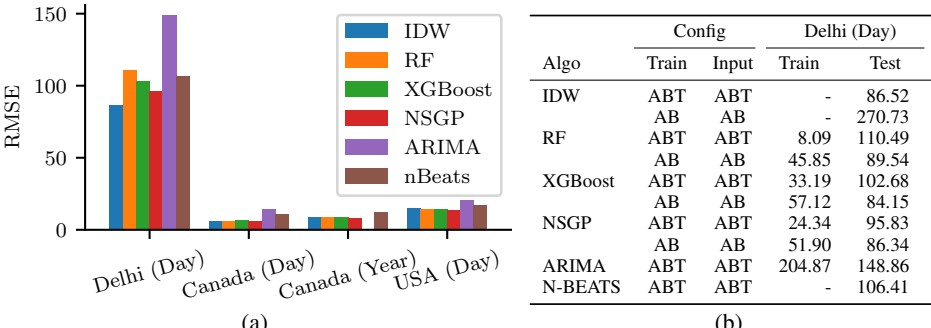

| Algo | Config | | Delhi (Day) | |
| | Train | Input | Train | Test |
|---|---|---|---|---|
| IDW | ABT | ABT | - | 86.52 |
| | AB | AB | - | 270.73 |
| RF | ABT | ABT | 8.09 | 110.49 |
| | AB | AB | 45.85 | 89.54 |
| XGBoost | ABT | ABT | 33.19 | 102.68 |
| | AB | AB | 57.12 | 84.15 |
| NSGP | ABT | ABT | 24.34 | 95.83 |
| | AB | AB | 51.90 | 86.34 |
| ARIMA | ABT | ABT | 204.87 | 148.86 |
| N-BEATS | ABT | ABT | - | 106.41 |

| (a) | (b) |

Figure 7: Forecasting RMSE. (a) Normalization is done across all days. (b) Comparison of normalization across all days (T) vs normalization over each day.

# 5   Limitations and Future Work

We acknowledge this dataset is limited to 3 months, and we are working to scale in another metropolitan city in India for a comprehensive 6+ months data collection and analysis. The bus routes selected by us covered the different types of areas, including green cover, residential, commercial etc, termed as static factors. Hence we have tried our best to reduce the bias towards any particular factor. Still, there is limited spatial distribution due to the bus routes over urban arterial roads. An alternative could have been to put sensors in private vehicles, but it is not reasonable to drive cars just to gather the data. Also, commercial cabs can be used for data collection, which we can check in our future work. We are anyhow moving from a high limitation of few data points in the same vicinity to million+ points.

In Appendix G, we also discuss anomaly detection used to fix any faults in the low cost sensors. We check for samples recorded per minute, number of minutes each device is active in an hour, number of active hours in a day, samples recorded per region, inter-sensor PM values variation, and intra-sensor PM values variation, during dataset collection to effectively keep validating the data being collected.

In our future work, we aim to address the problem of recommending suitable locations for installing new expensive sensors effectively within budget constraints, a challenging task in a developing country like India. By leveraging the insights gained from this research, we strive to optimize the allocation of resources and enhance the efficiency of the monitoring network, further strengthening pollution mitigation efforts.

# 6   Conclusion

Delhi-NCR, with its notorious air pollution problem, poses a significant health risk to its population of approximately 46 million individuals. In this paper, we present a novel PM dataset AIRDELHI collected from this region using low-cost IoT devices deployed on public buses. This dataset serves as a valuable resource for environmental researchers and medical practitioners, offering insights into ground-level PM exposure for daily commuters and temporal variations in PM levels over days and weeks. It provides a comprehensive view of spatial variations across different locations within the region.

Through thorough statistical analysis and benchmarking studies, we have established that the released dataset is distinct from any other existing pollution dataset. By comparing the performance of machine learning algorithms on the released dataset against the Canada and USA datasets, we have demonstrated the significant differences in characteristics and challenges associated with the Delhi-NCR dataset. This highlights the need for specialized approaches and tailored solutions to address the unique complexities of air pollution in this region.

The low-cost mobile data collection has the potential to complement the expensive static sensor network in the city, empowering citizens to make informed decisions regarding local PM levels. This includes determining the safety of engaging in outdoor activities, choosing appropriate protective measures such as face-masks or air purifiers, and selecting optimal commuting routes and transportation modes to minimize PM exposure. Such considerations are vital for safeguarding public health and promoting environmental sustainability.

To foster further advancements in the field of environmental sustainability, we release both the code and data associated with this study. This allows researchers to build upon our work, explore new avenues of inquiry, and contribute to the collective understanding and management of air pollution-related challenges.

## Acknowledgments and Disclosure of Funding

This work was supported by DST-SERB for funding through IMPRINT-II research grant. We also thank the Delhi Integrated Multimodal Transit System (DIMTS) for allowing us to instrument their fleet.

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
