# Appendix

## A    Graphsage (with Graph formulation)

We aim at learning universal weights, similar to GraphSAGE Hamilton *et al.* [2017], which will signify the importance of a neighbour based on some known node values and edge weights. Here we define node values as the value of the pollutant PM2.5 while the edges are created using latitude, longitude and datetime features. Firstly, a graph is created from the train dataset, aggregating all inputs within 500m and 30 minutes of each other into a single node. An edge is created between two nodes if they lie within 2 hours of each other. The graph then goes through two graph-based layers to learn the required weights where embeddings are learnt using the max and mean aggregation layers, followed by 3 fully connected neural network layers to predict the final pollutant value.

Let $G = (V, E, \sigma, \mathcal{A})$ be a Directed Graph with $V$ vertices/nodes, $E$ edges, $\mathcal{A}$ attributes and $\sigma$ as the label mapping, where

$\sigma : V \to \mathcal{L}$

$\mathcal{L}$ being the set of PM$_{2.5}$ values.

V corresponds to the spatiotemporal locations where PM$_{2.5}$ values are known (S: Red) or desired (U: Blue), i.e. V=S+U. E ($e \in E$) connects the V ($v \in V$) such that

$e_{ij} = (v_i, v_j) \mid v_i \in S \land v_j \in (S \lor U)$ and $t_{ij} \leq TimeLimit$, where $t_{ij}$=abs($v_i^t$ - $v_j^t$)

The Graph $G$ comprises of separate connected components for different days.

$e_{ij} = (v_i, v_j) \mid v_i \in Day_p$ and $v_j \in Day_q \Rightarrow p = q$

Weight of each edge is inversely proportional to the spatial distance between the two nodes across the edge.

$w_{ij} = \frac{1}{1+d_{ij}}$, if $e_{ij}$ exists, where $d_{ij}$=haversine($v_i, v_j$)

Edges exist from all S nodes to each U node. No S to S edges exist.

$e_{ij} = (v_i, v_j) \mid v_i \in S$ and $v_j \in U \Rightarrow |e_{ij} \, \forall i| = |S| \, \forall j$

The graph G is of two types:

**Train Graph** $G_{Train}$**:** It is used for training Graphsage Neural Network.

$v \in Day_{Train} \Rightarrow v \in S \lor U \Rightarrow |v \in S| > 0$ and $|v \in U| > 0$

The RMSE loss on the nodes $v \in U$ is used for model training.

**Test Graph** $G_{Test}$**:** It is used for evaluating the trained Graphsage model on unseen test day data ($Day_{Test}$) along with full data from known days.

$v \in Day_{Test} \Rightarrow v \in S \lor U \Rightarrow |v \in S| > 0$ and $|v \in U| > 0$

The $v$ is formed by taking the corresponding PM$_{2.5}$ label $L$ and an indicator variable $I$.

$v_i = L_i | I_i$

$L_i \leftarrow PM_{2.5}, I_i \leftarrow= 1 \, \forall \, v \in S$

$L_i \leftarrow 0, I_i \leftarrow= 0 \, \forall \, v \in U$

The 2 layer mean-pool and max-pool model graphsage architecture is shown in Fig. 8.

The RMSE loss of the nodes $v \in U$ (or $v \in P$ in particular) is used as the reporting metric.

For Graphsage based evaluation, out the 80% training data in 5-fold cross validation, we use 40% as *visible* set, 40% as *held-out* set, to manage edges between these two sets.

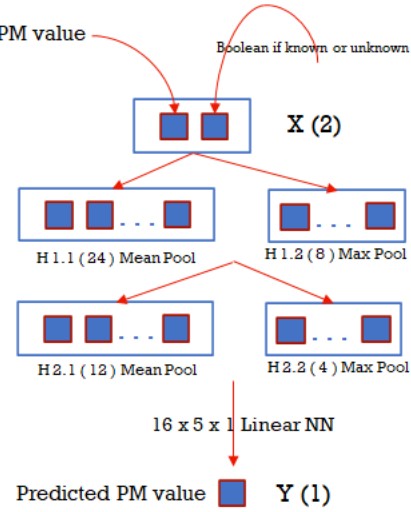

Figure 8: Graphsage model architecture.

# B   Complete ML Benchmarks

Table 3 shows the complete benchmark for Spatio-temporal Interpolation for different train and input configurations. An important subset of these benchmarks is presented in Fig. 5 and discussed in § 4.4 in the main paper.

Table 3: Spatiotemporal Interpolation RMSE for different configurations (* denotes partial experiments).

| Algo | Config | | Delhi (Day) | | Canada (Day) | | Canada (Year) | | USA (Day) | |
|------|--------|-------|-------------|------|--------------|------|---------------|------|-----------|------|
| | Train | Input | Mean | S.D. | Mean | S.D. | Mean | S.D. | Mean | S.D. |
| MeanPred | - | C | 65.80 | 2.44 | 3.13 | 1.14 | 5.66 | 1.13 | 13.85 | 3.02 |
| IDW | ACT | ACT | 39.94 | 2.51 | 2.56 | 0.95 | 4.56 | 1.05 | **10.24** | 2.57 |
| | AC | AC | 351.73 | 2.85 | 2.66 | 0.95 | 7.33 | 1.61 | 23.21 | 5.29 |
| | C | C | 25.83 | 2.77 | **2.31** | 0.98 | 4.35 | 0.91 | 10.32 | 2.60 |
| RF | ACT | ACT | **22.24** | 2.81 | 2.37 | 0.95 | 4.18 | 0.68 | 10.73 | 2.89 |
| | AC | AC | 77.30 | 2.67 | 2.69 | 0.98 | 6.05 | 0.93 | 13.93 | 3.20 |
| | C | C | 22.25 | 2.77 | 2.34 | 0.89 | 4.12 | 0.68 | 10.82 | 2.85 |
| XGBoost | ACT | ACT | 33.24 | 2.87 | 2.55 | 0.95 | 4.62 | 1.01 | 11.51 | 3.05 |
| | AC | AC | 65.04 | 2.55 | 2.90 | 0.98 | 6.03 | 0.84 | 14.19 | 3.32 |
| | C | C | 29.73 | 2.76 | 2.71 | 1.05 | **4.09** | 0.67 | 11.66 | 3.16 |
| NSGP | ACT | ACT | 29.11 | 3.84 | 2.57 | 1.09 | 4.41 | 0.89 | 10.39 | 2.69 |
| | ACT | C | 194.96 | 1.63 | 13.02 | 0.72 | 14.68 | 0.63 | 27.11 | 3.25 |
| | AC | AC | 69.75 | 3.65 | 2.89 | 0.90 | 5.99 | 0.95 | 13.42 | 3.09 |
| | AC | C | 37.46 | 4.63 | 3.17 | 1.12 | 5.25 | 1.22 | 22.14 | 3.46 |
| | C | C | 170.99 | 9.31 | 12.74 | 0.55 | 13.51 | 0.72 | 27.81 | 3.67 |
| Graphsage | AC | C | 38.63 | 3.89 | 2.96 | 1.25 | 5.37 | 1.13 | 11.66 | 3.29 |
| | C | C | 38.68 | 4.12 | 3.13 | 1.24 | 5.68 | 1.46 | 12.75 | 4.06 |

Table 4 shows the complete benchmark for Spatio-temporal Missing data Imputation for different train and input configurations. Missing data imputation is briefly discussed in § 4.4 in the main paper. The traditional and powerful RF (Random Forest) algorithm outperforms all other algorithms and methods.

Table 4: Missing Data Imputation RMSE for different configurations.

| Algo | Config | | Delhi (Day) | | Canada (Day) | | Canada (Year) | | USA (Day) | |
|---|---|---|---|---|---|---|---|---|---|---|
| | Train | Input | Mean | S.D. | Mean | S.D. | Mean | S.D. | Mean | S.D. |
| MeanPred | - | C | 65.80 | 2.44 | 3.13 | 1.14 | 5.66 | 1.13 | 13.85 | 3.02 |
| IDW | ABCT | ABCT | 40.06 | 2.51 | 2.56 | 0.95 | 4.56 | 1.05 | 10.19 | 2.57 |
| | ABC | ABC | 399.44 | 1.14 | 2.69 | 0.93 | 7.92 | 1.47 | 68.63 | 8.00 |
| RF | ABCT | ABCT | **22.26** | 2.85 | **2.34** | 0.93 | **4.22** | 0.67 | **9.42** | 2.60 |
| | ABC | ABC | 78.90 | 2.71 | 2.70 | 0.96 | 6.21 | 0.96 | 14.09 | 3.13 |
| XGBoost | ABCT | ABCT | 33.46 | 2.87 | 2.53 | 0.91 | 4.63 | 1.02 | 10.23 | 2.74 |
| | ABC | ABC | 67.66 | 2.55 | 2.94 | 0.96 | 6.19 | 0.87 | 13.84 | 3.12 |
| NSGP | ABCT | ABCT | 29.06 | 3.64 | 2.52 | 0.95 | 4.40 | 0.85 | 9.62 | 2.46 |
| | ABC | ABC | 71.27 | 3.16 | 2.81 | 0.91 | 6.09 | 0.88 | 13.38 | 2.97 |
| | ABC | C | 171.94 | 8.08 | 12.71 | 0.53 | 13.29 | 0.94 | 21.76 | 3.18 |
| | ABCT | C | 194.98 | 1.55 | 12.90 | 0.60 | 14.58 | 0.68 | 26.80 | 3.08 |
| | ABT | C | 195.86 | 3.00 | 13.03 | 0.61 | 14.68 | 0.95 | 27.28 | 2.99 |
| | AB | C | 37.63 | 3.87 | 4.15 | 0.92 | 5.43 | 1.09 | 23.19 | 3.10 |
| Graphsage | ABC | C | 38.53 | 2.94 | 3.15 | 1.30 | 5.46 | 1.11 | 11.78 | 3.56 |
| | AB | C | 38.48 | 2.86 | 3.13 | 1.25 | 5.41 | 1.08 | 11.59 | 3.15 |

Table 5 shows the complete benchmark for Spatio-temporal Forecasting for different configurations. A subset of these benchmarks is presented in Fig. 7 and discussed in § 4.4 in the main paper.

Table 5: Forecasting RMSE for different configurations.

| Algo | Config | Delhi (Day) | Canada (Day) | Canada (Year) | USA (Day) |
|---|---|---|---|---|---|
| IDW | ABT | 86.52 | 5.65 | 8.31 | 14.61 |
| | AB | 270.73 | **5.73** | 11.23 | 69.20 |
| RF | ABT | 110.49 | 5.90 | 8.45 | 14.23 |
| | AB | 89.54 | 6.11 | 10.80 | 14.58 |
| XGBoost | ABT | 102.68 | 6.69 | 8.23 | 14.25 |
| | AB | **84.15** | 6.51 | 9.84 | 14.52 |
| NSGP | ABT | 95.83 | 5.76 | **8.01** | **13.65** |
| | AB | 86.34 | 6.08 | 10.22 | 14.34 |
| ARIMA | ABT | 148.86 | 13.87 | 12.85 | 20.12 |
| nBeats | ABT | 106.41 | 10.88 | 11.84 | 17.05 |

## NSGP Variance

Non-stationary GP models provides us with uncertainty (variance) values around the expected mean PM2.5 value for each expected spatio-temporal location. We find that the average variance value for Delhi dataset is huge as compared to Canada (Day) experiments. It is more challenging for a model or algorithm to correctly understand and predict the PM values for Delhi dataset. Even the USA dataset with data over a big region does not exhibit such complexity for the algorithms.

Table 6: NSGP Variance.

| | Delhi (Day) | Canada (Day) | Canada (Year) | USA (Day) |
|---|---|---|---|---|
| Spatio-temporal Interpolation | 118.73 | 17.29 | 72.94 | 76.34 |
| Missing Data Imputation | 142.51 | 20.34 | 113.37 | 72.58 |
| Forecasting | 77.38 | 19.96 | 60.89 | 59.76 |

## C  Anova Tests Analysis for Low Cost Sensor

In continuation to the data quality analysis presented in § 3.2, we performed Anova Tests over the data collected by DustTrak and our Low Cost Mobile sensor devices at the same location. ANOVA Navidi [2009], Analysis of Variance, is a strong statistical factorial technique which involves one dependent variable known as response variable and one or more independent variables known as factors. The factors have different levels called treatments. The ANOVA tests compare two types of variation, the variation between the sample means and the variation within the samples.

**Two-way ANOVA test between DustTrak reference sensor and our low-cost mobile sensor**

In relation to our low cost sensor scenario, the observed $PM_{2.5}$ values are dependent on the sensor *Type* (DustTrak vs Low Cost) and the time(*Day*) of observation. As we have two factors, we need to perform two-way ANOVA test. For the *Day* factor, we take the hourly $PM_{2.5}$ mean samples grouped over each day (24 hours) of observations.

**Two-way ANOVA tests three *null* hypotheses**

    (a)  the means of observations grouped by factor *Type* are same

    (b)  the means of observations grouped by factor *Day* are same

    (c)  there is no interaction between the two factors *Type* and *Day*

**Two-way ANOVA Assumptions**

We make the standard assumptions of completeness, balanced design, normal distribution, similar variance, and sufficient replicates per treatment for validating ANOVA hypotheses. We take one device per sensor *Type* and same number (11) of *Day* as treatments under the two factors, with each *Type* and *Day* containing $PM_{2.5}$ samples. Fig. 9 shows the box-plot diagram with similar standard deviation for the DustTrak and our Low cost mobile sensors.

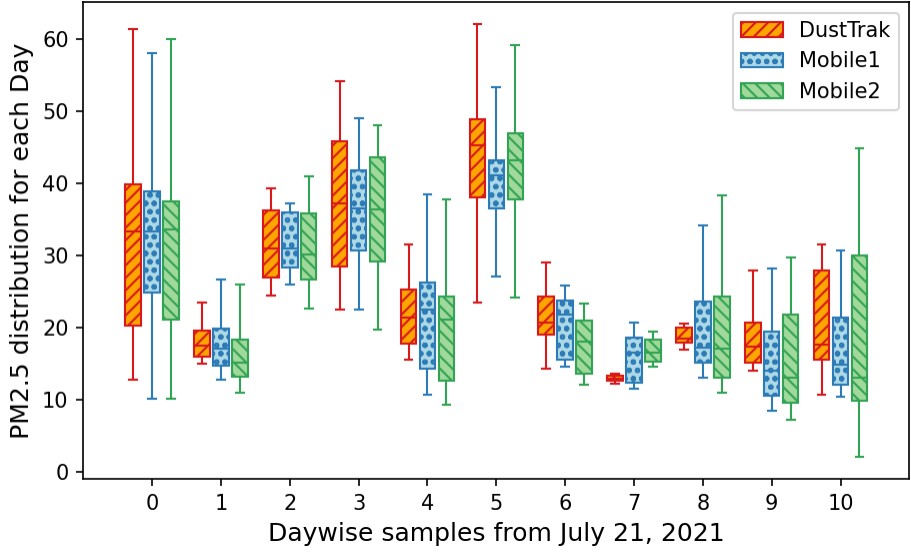

Figure 9: Mean and Standard Deviation for DustTrak and our Low Cost Mobile sensors.

**Interpreting two-way ANOVA results**

Table 7 shows the two-way ANOVA test results for DustTrak and our Low Cost Mobile sensor. As per Seltman [2018], the *SumSq* column represents the sum of squared deviations for each *Source* of variation. Each *Source* has a *df* (degrees of freedom) which is a measure of the number of independent pieces of information present in the deviations that are used to compute the corresponding *SumSq*. Each *MeanSq* is a variance estimate and the *SumSq* divided by the *df* for that *Source*.

Each *F*-statistic is the ratio of two *MeanSq* values. For the main effects, *Type* and *Day*, the denominators are all MSE which are pure estimates of group variance, unaffected by the validity of the null hypothesis. Each *F*-statistic is compared against it's null sampling distribution to compute a *p-value*. Interpretation of each of the *p-values* depends on the corresponding null hypothesis.

Table 7: Two-way ANOVA test for DustTrak Reference Sensor vs Our Low Cost Sensor Mobile Sensor 1

| Effect | Source | df | SumSq | MeanSq | F | p-value | Significance |
|---|---|---|---|---|---|---|---|
| Main | *Type* | 1 | 197.84 | 197.84 | 2.36 | 0.1248 | Holds hypo (a) |
| | *Day* | 10 | 30204.98 | 3020.50 | 36.10 | < 0.0001 | Reject hypo (b) |
| Interaction | *Type*Day* | 10 | 261.76 | 26.18 | 0.31 | 0.9778 | Holds hypo (c) |
| Error | Residual | 444 | 37147.11 | 83.66 | | | |

In the presence of an interaction (*Type*Day*), the *p-value* for the interaction is most important and the main effects *Type* and *Day* p-values would be ignored if the interaction is significant. This is mainly because if the interaction is significant, then some changes in both explanatory variables (*Type* and *Day*) must have an effect on the outcome $PM_{2.5}$, regardless of the main effect *p-values*. The null hypothesis for the interaction *F*-statistic supports an additive relationship between the two explanatory variables, *Type* and *Day*, in their effects on the outcome $PM_{2.5}$. If the *p-value* for the interaction is less than $\alpha$ (usually 0.05), then we have a statistically significant interaction.

As we have a non-significant interaction $F_{1,10} = 0.31$ with *p-value* = 0.9778 which is greater than $\alpha = 0.05$, the null hypothesis (c) holds and the *p-values* for the main effects are valid for consideration. So, we can see that the *Day* has a significant *p-value* and thus it rejects the null hypothesis (b) meaning that there is impact of different *Day*'s observation on the observed $PM_{2.5}$ sample. This outcome aligns with a common understanding regarding the varying pollution across different days.

The analysis for the main effect sensor *Type* is more encouraging. It has a non-significant *p-value* = 0.1248 which holds the null hypothesis (a) that the means of the observations of the two device *Types*, DustTrak and our Low Cost Mobile sensor, are same. Hence, our Low Cost Mobile device can be effectively used to collect $PM_{2.5}$ observations in place of the expensive DustTrak sensors.

**One-way ANOVA test between DustTrak reference sensor and our low-cost mobile sensor**

Though the two-way ANOVA results hold for the main effects, we still perform one-way ANOVA test for the main effect *Type* (DustTrak vs Low Cost) for the observed $PM_{2.5}$ values. We ignore the *Day* factor in this analysis, so the $PM_{2.5}$ samples are only attributed with the *Type* factor. One-way ANOVA tests for the hypothesis (a) as of two-way ANOVA and with the standard assumptions of normal distribution and similar variance.

Table 8: One-way ANOVA test for DustTrak Reference Sensor vs Our Low Cost Sensor Mobile Sensor 1

| Effect | Source | df | SumSq | MeanSq | F | p-value | Significance |
|---|---|---|---|---|---|---|---|
| Main | *Type* | 1 | 197.84 | 197.84 | 1.36 | 0.2445 | Holds hypothesis (a) |
| Error | Residual | 464 | 67613.85 | 145.72 | | | |

Table 8 presents the results for one-way ANOVA, which too shows *Type* factor to have a non-significant *p-value* = 0.2445 which holds the null hypothesis (a). Hence with similar means of the observations, our Low Cost Mobile device can replace the expensive DustTrak sensors.

**Two-way ANOVA test for our Low Cost device replaceability**

We also show that our Low Cost Mobile devices are replaceable by each other. We perform two-way ANOVA tests between our Low Cost Mobile devices and the results are presented in Table 9.

As the *p-value* for the interaction is non-significant, main effects are valid. Likewise *Day* factor rejects hypothesis (b) and importantly *Type* factor holds hypothesis (a), allowing our Low Cost devices to replace each other as applicable.

Table 9: Two-way ANOVA test for Our Low Cost Sensor Mobile Sensor 1 vs 2

| Effect | Source | df | SumSq | MeanSq | F | p-value | Significance |
|---|---|---|---|---|---|---|---|
| Main | *Type* | 1 | 145.65 | 145.65 | 1.65 | 0.1991 | Holds hypothesis (a) |
| | *Day* | 10 | 31204.66 | 3120.47 | 35.43 | < 0.0001 | Reject hypothesis (b) |
| Interaction | *Type*Day* | 10 | 148.46 | 14.85 | 0.17 | 0.9982 | Holds hypothesis (c) |
| Error | Residual | 450 | 39632.11 | 88.07 | | | |

# D   Spatio-temporal Correlation and Covariance Analysis

To analyze the temporal correlation over the $PM_{2.5}$ values for different locations, we split the data in grids of 1km x 1km x 1hr and average the $PM_{2.5}$ values in each grid to get a representative value[1].

We observe high autocorrelation for different spatial grid locations, for 1 hour lag. In Fig. 10, we show the autocorrelation for 12 spatial grid locations, denoted as locations A-L, with the corresponding Latitude-Longitude marked alongside in the titles. The X and Y axis represent the $PM_{2.5}$ values for without lag (y(t)) and with 1 hour lag (y(t+1)) respectively.

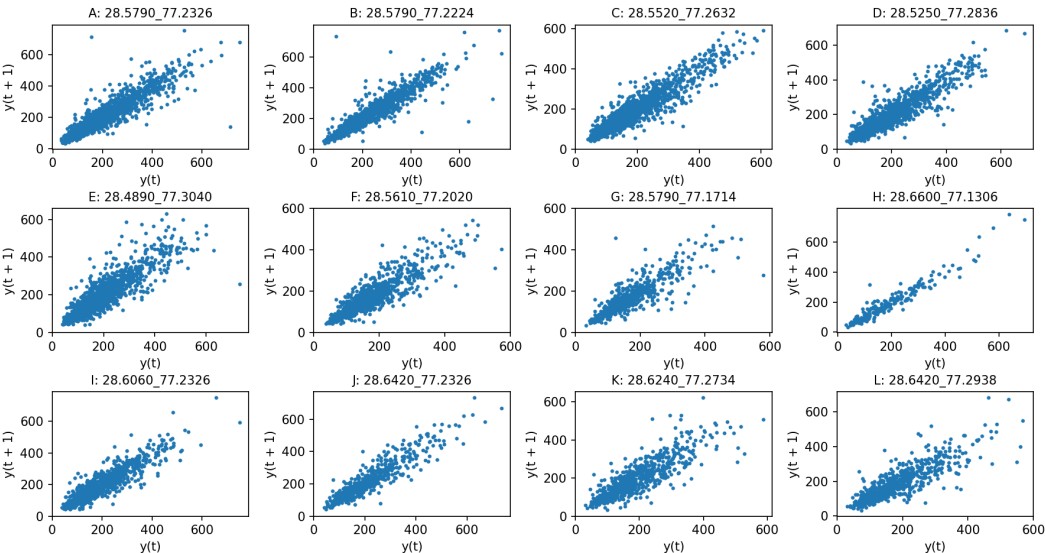

Figure 10: autocorrelation for 12 grid locations for 1 hour lag (the titles contain the latitude-longitude of the grid locations).

The autocorrelation decreases for lags of 2 hour or more at individual grid locations.

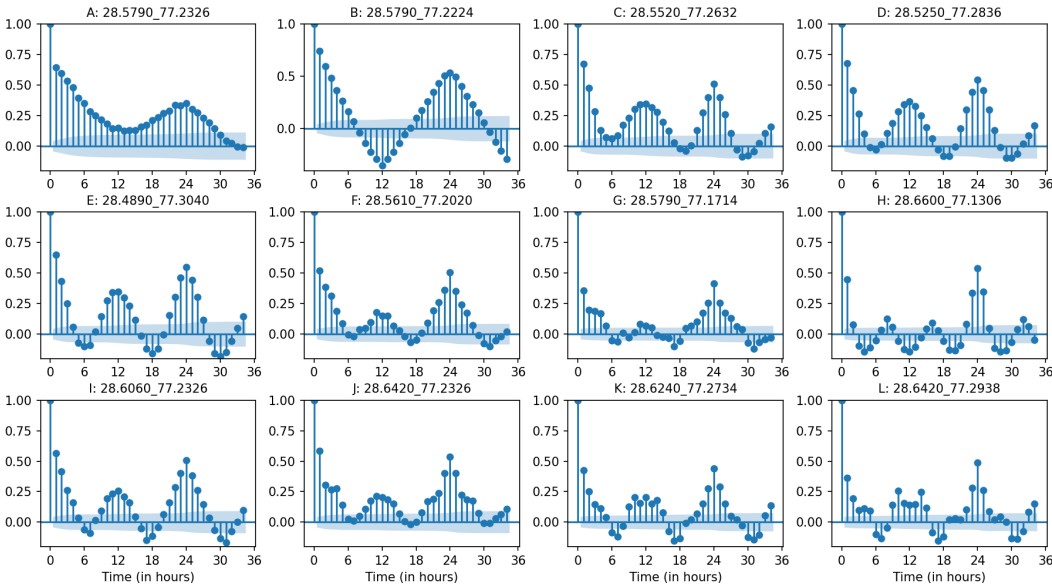

Figure 11: autocorrelation for 12 grid locations up to 1.5 days lags.

---

[1]This dataset version is available at `https://huggingface.co/datasets/sachin-iitd/DelhiPollDataset/tree/main/4.Grid(PM+Met)`

We further analyze the hourly data for 1.5 days for individual locations, and observe patterns of high and low autocorrelation. Fig. 11 shows autocorrelation for the same 12 grid locations for 36 hours. We observe a high autocorrelation at 24 hour period indicating similar pollution traits at the same time next day. We further observe that most locations (except A and B) exhibit a local high autocorrelation in the sub 12 or 8 hour periods as well, indicating repeated traffic patterns, like similar pollution characteristics around the morning and evening periods.

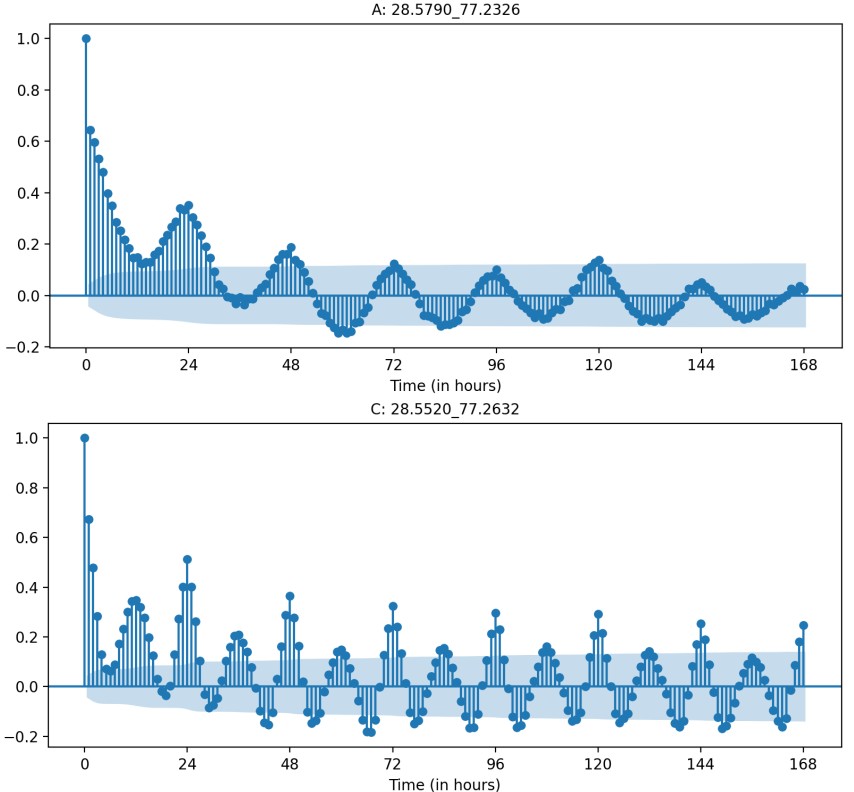

Figure 12: autocorrelation for A and C grid locations for up to 7 days lags. The autocorrelation decreases with increasing the lag.

The locations A and B seem to exhibit behaviour different from other locations, with low autocorrelation in the sub 12 hours period. Both these locations are adjacent to *Kushak Nalla Bus Depot* which hosts the buses for stopping between the runs and for overnight stopping while the bus services are down. Hence more data is collected here for longer periods which shows different pollution traits compared to all other grid locations. For grid locations A and C, Fig. 12 shows the autocorrelation for the two distinct behaviour for 7 days lags. The autocorrelation for 7 days for grid locations A and C is also presented as heat-map in Fig. 13.

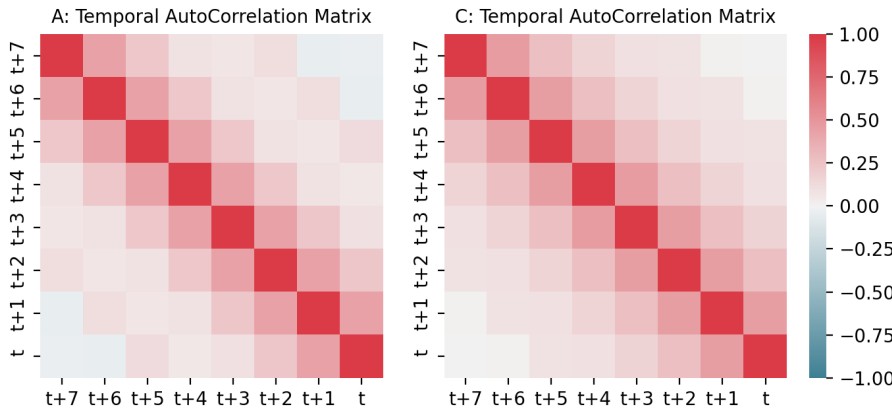

Figure 13: autocorrelation for A and C grid locations for up to 7 days lags as heat-map. The autocorrelation decreases with increasing the lag.

We also analyze the covariance among the spatial grid locations. In Fig. 14, we observe low or different covariances for locations at same distance from the base location. Hence there are different pollution traits at different locations, which sometimes match with neighboring locations due to common local/global factors, and sometimes differ due to some local factors.

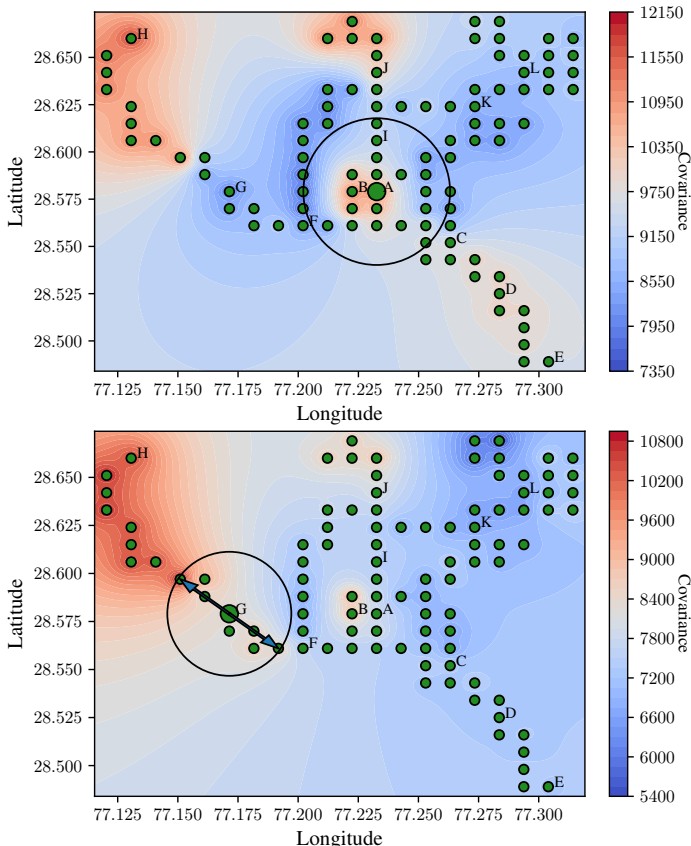

Figure 14: Interpolated empirical covariance between a base locations and other (99) grid locations in Delhi dataset. In the first plot, all the locations around the base location A has very low covariance with A. In the second plot, the locations at same distance from the base location G have different covariance w.r.t. G.

## E    Meteorological Factors (Temperature, Relative Humidity, Wind Speed) Analysis

A recent research Yang *et al.* [2020] focuses the effect of meteorological factors on the pollution traits in Shenyang, China to understand temporal-spatial characteristics of particles and analyze the causality factors. Similarly, we analyzed the given dataset for the impact of the collected meteorological parameters - Temperature and Relative Humidity (RH).

We also purchased Wind Speed (WS) data for the 3 months (Nov 2020 - Jan 2021) from `www.windfinder.com`, and received data with half-hourly frequency for IGI Airport. Data has wind speed in knots varying from 0 to 19, with maximum daily average being ~7 knots. This wind data is not very fine, it has compromised precision with integer values only with some values missing. Still we could use this to analyze the effect of wind speed on the pollution.

Table 10: Correlation of Temperature, RH and WS with PM values.

|  | Temperature | Relative Humidity (RH) | Wind Speed (WS) |
|---|---|---|---|
| $PM_1$ | -0.305 | 0.323 | -0.508 |
| $PM_{2.5}$ | -0.303 | 0.332 | -0.480 |
| $PM_{10}$ | -0.317 | 0.335 | -0.477 |

Table 10 shows the correlation of temperature, humidity and WS with PM values. Overall we observe a negative correlation of PM with temperature, positive correlation of PM with RH, and negative correlation of PM with WS. Other researchers, like Yang *et al.* [2020]; Liu *et al.* [2020]; Avdakovic *et al.* [2016], have also observed similar positive and negative correlations, which is intuitive as temperature expands the air reducing PM per unit volume, humidity/moisture intensifies it, and wind blows it away. However, at a given time, which meteorological factor among these three will dominate needs more complex ML modeling.

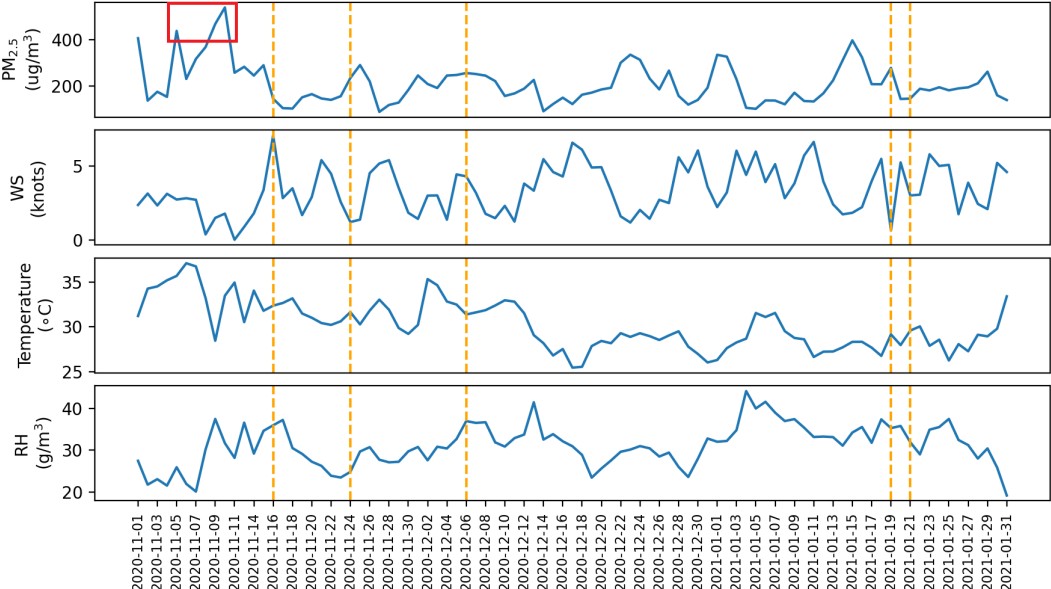

Figure 15: Meteorological factors vs PM$_{2.5}$ over the 3 months. The orange vertical lines show the relation of high or low PM$_{2.5}$ with WS, temperature and RH. The red box shows the situation of very high PM$_{2.5}$ which is explained with non-meteorological causes.

Fig.15 shows the average wind speed, temperature and humidity for the 3 months with the corresponding average PM$_{2.5}$ values. We observe high WS with low PM on Nov 16 and low WS with high PM on Nov 24 and Jan 19, which matches the intuition. But there are adverse situations of high WS with high PM on Dec 6, and low WS with low PM on Jan 21 which doesn't match the intuition.

The initial days of November show very high pollution spikes with stable winds. As per Weather [2020], there was *high moisture, calm winds and stubble burning around the beginning of November 2020*. As per HT [2020], in 2020, Delhi had six consecutive *severe* days from November 5-10, the longest *severe* spell seen in the city since 2016. A combination of multiple factors affected this, including a prolonged and intensive stubble burning season that started early on and in high incidence, the firecracker bursting festival and unfavourable meteorological conditions.

Due to such severe pollution situation in Delhi-NCR around winters, CAQM [2022] revised their action plan for **(a)** *Very Poor* Air Quality to avoid dust generating construction activities during months of October to January, and **(b)** *Severe* Air Quality to instruct individual house owners to provide electric heaters to security staff to avoid open burning.

Besides the meteorological parameters provided with the dataset, external data like below can be useful while modeling -

1. **Meteorological data from ERA5:** `https://cds.climate.copernicus.eu/cdsapp#!/dataset/reanalysis-era5-single-levels`

2. **NASA Fire count (VIIRS):** `https://firms.modaps.eosdis.nasa.gov/active_fire`

3. **Pollution Data from other sources:** OpenAQ: (`openaq.org`), CPCB: (`cpcb.nic.in`)

4. **Photochemical modelling:** `www.camx.com`

5. **Planetary boundary layer height:** `www.nrsc.gov.in/readmore_atmosphere_planet`

6. **Traffic Data from the dataset duration:** `delhi-trafficdensity-dataset.github.io`

7. **ClimateLearn: state-of-the-art climate-data/ML-models framework:** Nguyen *et al.* [2023b]

8. **ClimaX: flexible and generalizable weather/climate deep learning model:** Nguyen *et al.* [2023a]

# F Bus Route Analysis

We analyzed the Delhi dataset based on the different (13) bus routes available in the data. This is shown in Fig. 16 with the overall PM$_{2.5}$ colour-map.

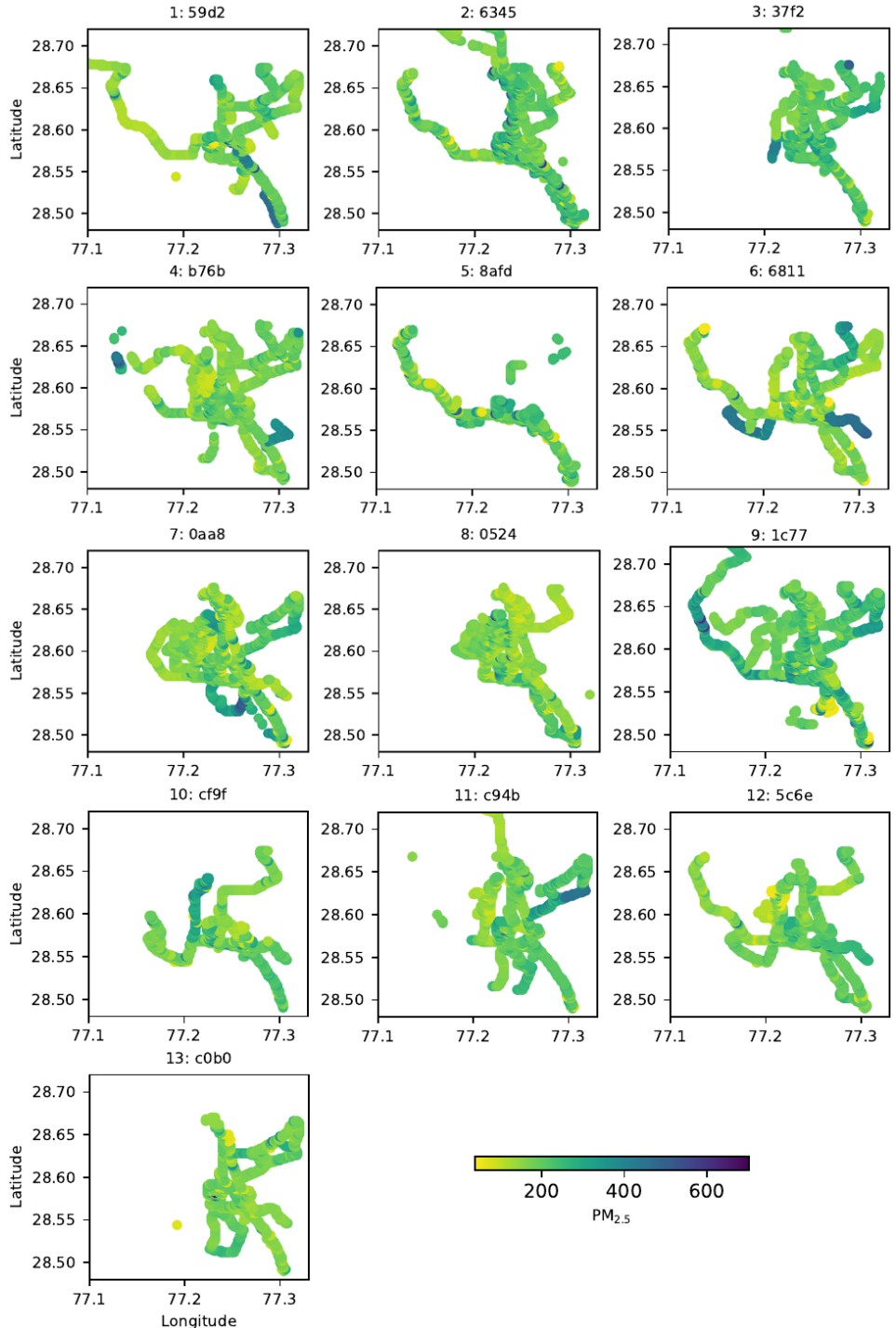

Figure 16: PM$_{2.5}$ distribution for the bus-routes followed over 3 months.

The title of each subplot shows the last 4 characters of the bus-route number from the dataset, with Latitude on Y-axis and Longitude on the X-axis. In different routes, there are sub-routes which showed high PM$_{2.5}$ concentrations in this duration, which can be due to high traffic concentrations

and or other local factors. Such route level $PM_{2.5}$ concentrations can be used to recommend people which routes to choose for commute/travel. Such and many other pollution traits can be further analyzed by using our dataset in conjunction with other open source data.

We further checked the daily $PM_{2.5}$ concentrations across the 13 routes. Fig. 17 shows the mean $PM_{2.5}$ values per day, with $PM_{2.5}$ levels on the Y-axis and day number on the X-axis. The number on the right axis in each subplot denotes the bus-route number.

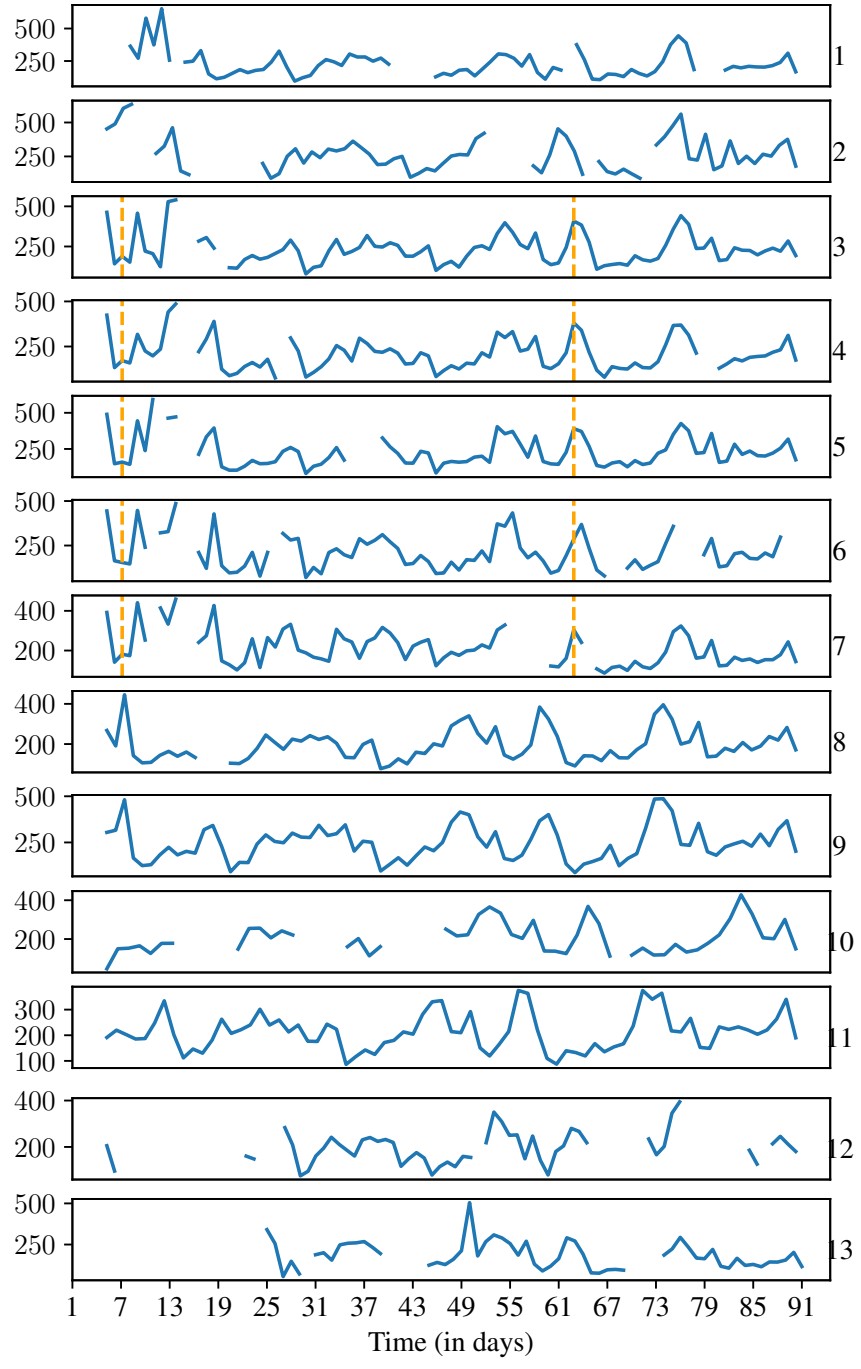

Figure 17: Daily $PM_{2.5}$ averages (denoted on left axis) for the different routes (mentioned on right axis) for the 3 months (x-axis). We observe some common low and high pollution periods, due to sharing of sub-routes by the buses.

As observed around the vertical lines in orange colour in Fig. 17, the different bus-routes seem to exhibit similar pollution traits due to the common paths tracked by them. We also observe the difference among different routes and hence a formal correlation analysis was performed to get better characteristics on the behaviour.

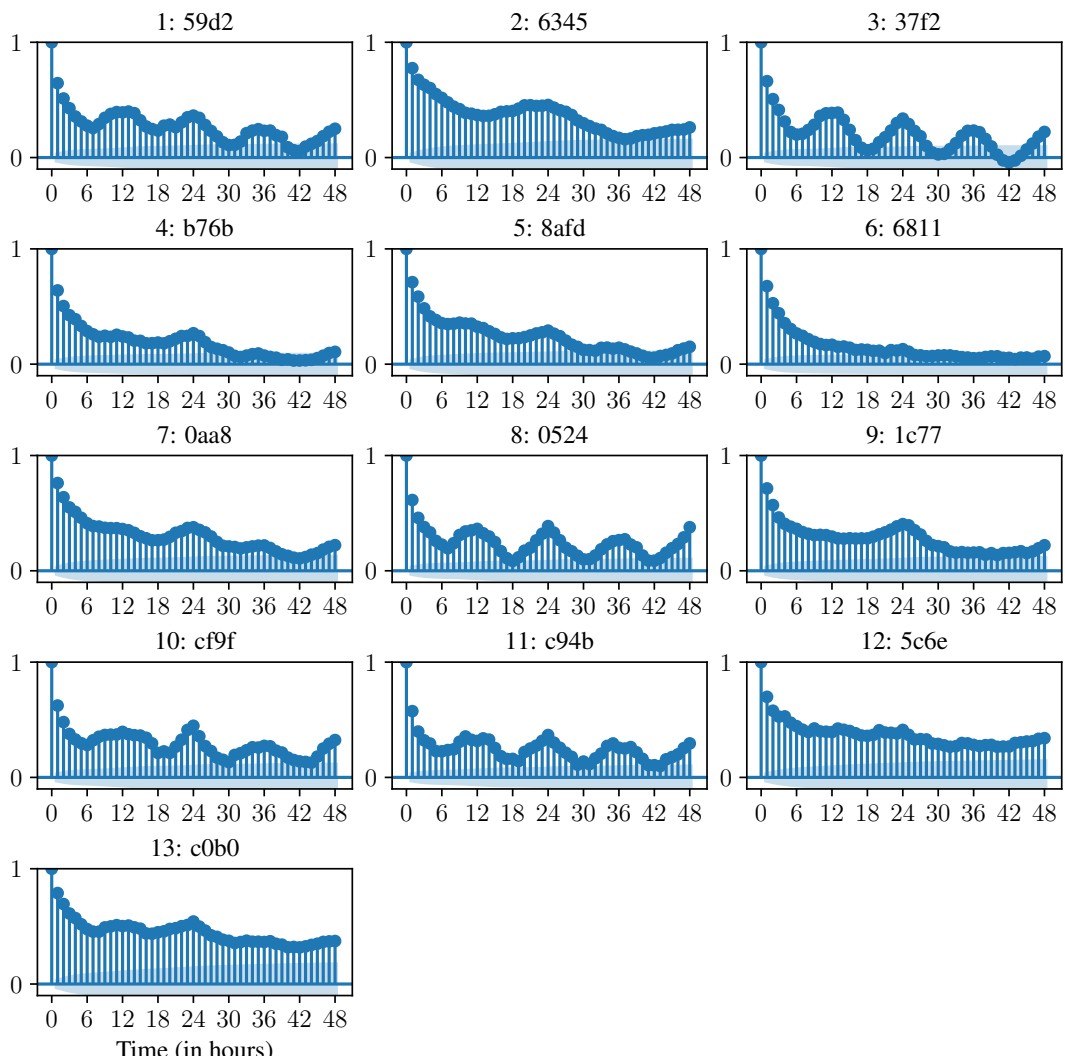

Figure 18: autocorrelation for the 13 bus-routes, for upto 2 days (48 hours) lags over hourly PM$_{2.5}$ averages. We observe positive autocorrelation in the bus-routes.

We performed the autocorrelation for the 13 bus-routes separately for hourly PM$_{2.5}$ averages. Similar to the combined route analysis done in Appendix D, we observe autocorrelation for separate bus-routes for 24 hour lags. As seen in Fig. 18, the level of autocorrelation is not same for all bus-routes, highlighting influence of local factors affecting pollution in their transit paths. Similar to overall correlation in Appendix D, we also find correlations at sub 12 hour lags as well for some of the bus-routes. autocorrelation for the 3 types of bus-routes, for upto 7 days lags over hourly PM$_{2.5}$ averages is shown in Fig. 19. We observe both positive and negative correlation in bus-route 3, almost no long-term correlation in bus-route 6 and varying but positive correlation in bus-route 11.

*In contrast to the grid level autocorrelation analysis performed in Appendix D, we observed less negative autocorrelation for most of the bus-routes, which seems due to the same paths being traversed at same time each day by the buses.*

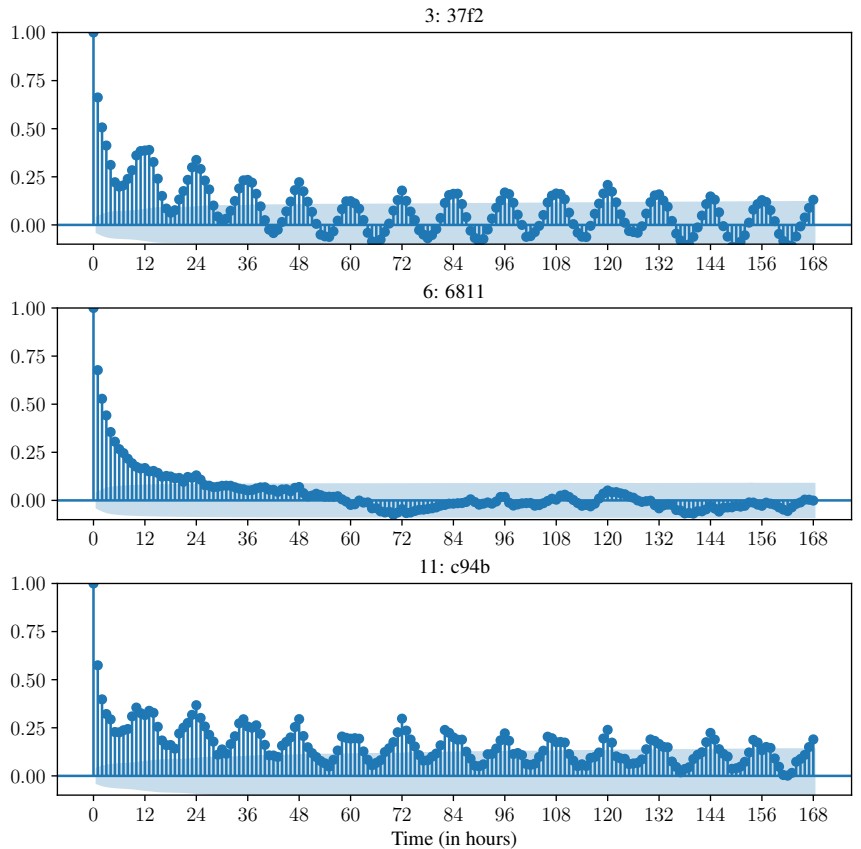

Figure 19: autocorrelation for the 3 types of bus-routes, for upto 7 days lags over hourly PM$_{2.5}$ averages. Both positive and negative correlation in bus-route 3, almost no long-term correlation in bus-route 6, varying but positive correlation in bus-route 11.

## Routes Covariance Matrix

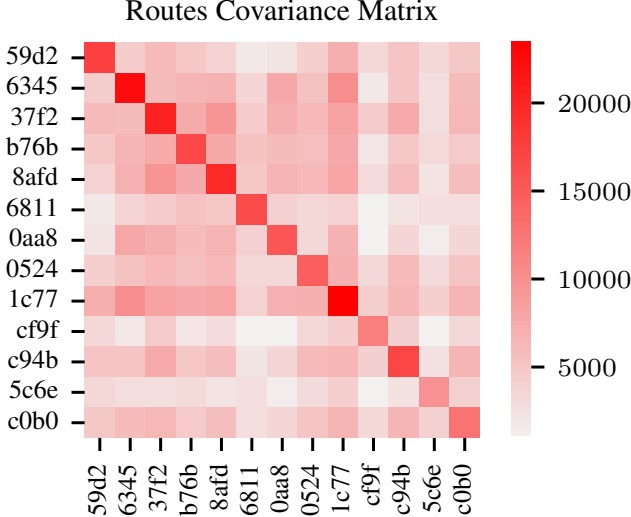

Figure 20: Buses share sub-routes hence some level of correlation exists between different bus routes.

Encouraged with the autocorrelation among different routes, we check the covariance among them, taking hourly PM$_{2.5}$ averages similar to autocorrelation. In Fig. 20, we can observe some average covariance among the different routes, while some of them show very low covariance. As Buses transit through different areas of the city, their corresponding data may contain peculiar characteristics for the area. A thorough analysis to understand those peculiar characteristics in contrast with other areas can reveal significant spatial traits for the Delhi region.

# G   Anomaly Detection

This dataset has been created using a novel IoT network with low cost sensor platform, deployed in public buses in a developing country, the first of its kind. There are many points of faults — sensors can be faulty, internet connection can be shaky, buses might be down .... the faults can affect the quantity of data as well as quality. Detecting such anomalies for quick fixes is a necessity. We applied statistical analysis to detect anomalies, which involves many heuristics with manually tuned thresholds. Our findings can serve as anomaly ground truth for this dataset. Automating this process with ML based methods (instead of manually tuned thresholds) can open up new avenues of anomaly detection in mobile and IoT networks. ML researchers can try and automate the fault detection process using our dataset and the ground truth anomalies. They can also modify our released code, to change our empirical thresholds for more or less aggressive anomaly definition. Additionally unsupervised learning methods perhaps would change the anomalies detected manually by us. We describe next the different anomaly metrics that we compute on the dataset using statistical analysis and empirically determined thresholds.

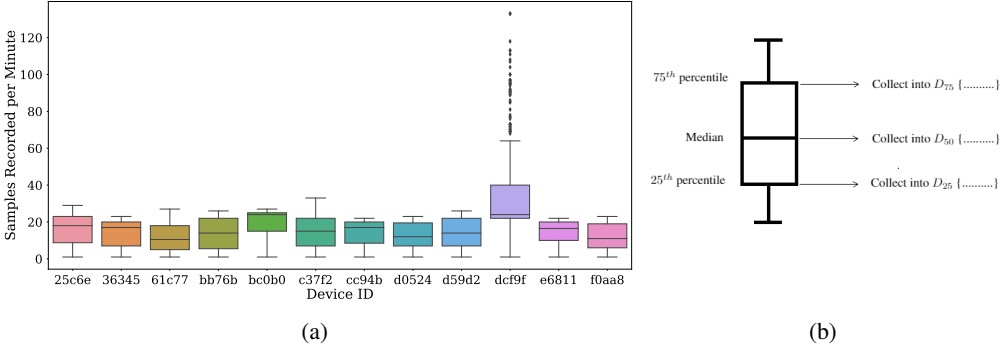

(a)          (b)

Figure 21: (a) Sampling rate i.e. number of samples recorded per minute for 3.12.2020. This helps us in finding out if any device isn't sampling properly. (b) Illustration of a sample box plot and process of collecting median, $25^{th}$ percentile and $75^{th}$ percentile in metric 1 and 2.

**Anomaly metric 1: Samples recorded per minute:** This metric checks for faulty devices which might be sampling more or less than expected rate. Fig. 21a shows ideal samples collected per minute should be around 20. If it deviates too much, that device is anomalous. The amount of deviation allowed is calculated statistically by observing the distributions for several days. Our algorithm (detailed in the supplementary section) finds the upper bounds and lower bounds of the median ($\Theta_{50}^L$, $\Theta_{50}^U$ ), $25^{th}$ percentile ($\Theta_{25}^L$, $\Theta_{25}^U$) and $75^{th}$ percentile ($\Theta_{75}^L$, $\Theta_{75}^U$) of the expected distribution. Anomaly is reported if any two of the three bounds are violated.

**Anomaly metric 2: Number of minutes each device is active in an hour:** A device can be active for all the 60 minutes of an hour or less based on time of the day/lunch break, stoppage at bus depots etc. So again we tried plotting the box plots for the distributions across the days and devices. We observe that ideally device should be active for 60 minutes of an hour, if the bus was taking a trip in that hour. So we used the same technique used in Metric 1: find the upper bounds and lower bounds of the median ($\Theta_{50}^L$, $\Theta_{50}^U$), $25^{th}$ percentile ($\Theta_{25}^L$, $\Theta_{25}^U$) and $75^{th}$ percentile ($\Theta_{75}^L$, $\Theta_{75}^U$) of the expected distribution. Anomaly is reported if any two of the three bounds are violated.

**Anomaly metric 3 : Number of active hours in a day:** An active hour for a particular sensor is any hour in which the sensor sends at least a fixed number($\gamma$) of samples. The number of active hours should ideally be greater than a threshold value($\tau$). But it is hard to fix one $\tau$ across all sensors, as different buses have different schedules and frequencies, which also can change over time. This is shown by the bar plots of number of active hours in Fig 22.

So, we define $\tau$ or ideal number of active hours for every sensor as the maximum of 10 active hours and $15^{th}$ percentile of the sensor's previous 15 days' active hours. This ensures that $\tau$ is appropriately chosen for every sensor depending on its particular bus's recent schedule. If the number of active hours for a sensor on any day doesn't satisfy the threshold ($\tau$), it is reported as anomalous for the day.

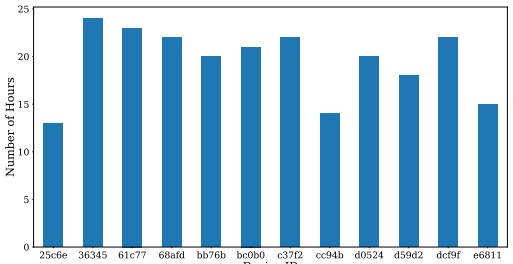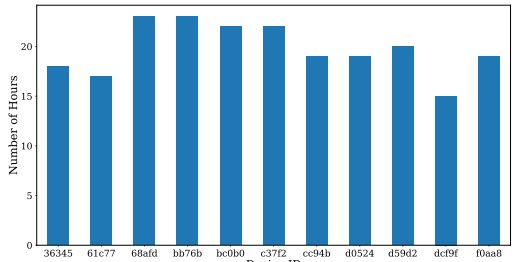

Figure 22: Number of active hours for different devices on two different days as shown by two different plots. We can see that the number of active hours on both days for each device are different.

**Anomaly metric 4: Samples recorded per region:** It is important to check whether daily around same number of data points are collected or not in an area. This metric detects situations where bus may not complete its scheduled trip due to mechanical breakdowns or high traffic resulting in less recording of data points in some areas. We divided the area covered by buses into 16 square regions. Given total number of data points collected in each region on a day, a region is reported as anomalous if its value deviates from past seven days average of that region by at least $\delta\%$. The value of $\delta$ is calculated by observing the data of several days.

**Anomaly metric 5: Inter-sensor PM values variation:** Ideally the PM values measured by different sensors should lie in a close range if the measurements were carried out at the same location and time. Every night from 0 AM IST to 5 AM IST all the buses remain parked at the same bus depot. We have used the PM value data from this time period to find devices whose PM 2.5 value measurements deviate from the general PM value trend of majority devices. For each hour, we have a box plot describing PM value distributions of all the devices as shown in Fig. 23a. Let $\Theta_{25}$ and $\Theta_{75}$ represents 25th and 75th percentile of distribution of PM values of a device during an hour. Interquartile range (IQR) is defined as $(\Theta_{75}-\Theta_{25})$. Given a box plot, a device is flagged for possible anomalous behaviour if it's IQR is very high (e.g. device e6811 in Fig. 23a). In order to define how much IQR should be considered high to be flagged, we define a threshold max_IQR which is set as 90 percentile of all the IQRs of all the devices in training set. Secondly, if a box (middle 50% data) of a device in the plot varies in a range different than the range of other devices then also the device is flagged (e.g. device bc0b0 in Fig 23a). To find such anomalies, we first find a range in which boxes of majority devices lie, then all those devices which are out of this range are flagged. A parameter called 'buffer' is defined statistically based on training data for finding the range. Given a PM value distribution of an hour, we iteratively calculate candidate range as $[\Theta_{25}$-buffer, $\Theta_{75}$+buffer] for each device. The candidate range which contains the boxes of maximum number of devices is considered as the final range. The devices whose box does not fit completely in this range are flagged for that hour. Finally a device is reported as anomalous if its get flagged for at least three hours in a day.

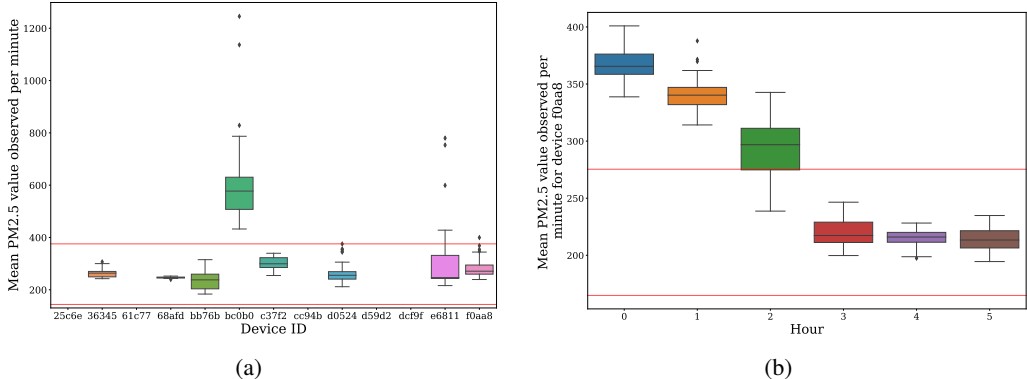

(a)                                        (b)

Figure 23: (a) shows a box plot of PM value distributions of various devices on 2020-12-20 4:00 AM IST. Red lines indicate the majority PM range. Device bc0b0 is flagged as it is out of majority PM range while device e6811 is flagged as its IQR exceeds max_IQR. They will be declared as anomaly if they show this behaviour in at least two more hours. (b) shows PM value distributions of the device f0aa8 on 2021-01-16. The device is reported as anomaly as its PM value measurements are highly varying across several hours.

**Anomaly metric 6: Intra-sensor PM values variation:** Similar to the above metric, intra sensor analysis verifies that the variation in a device's PM value recordings across consecutive hours is not very high. Given a device's PM value recordings during 0 AM IST to 5 AM IST, it is flagged for further checks if IQR of the device during any hour is greater than max_IQR or if at least three boxes lie out of majority PM range. The majority PM range is the PM value range which contains the maximum number of boxes computed similarly as described in the above metric except that the value of buffer here is computed based on intra senor PM value distributions. Finally a device is reported as anomalous only if its get flagged for at least three hours in a day. Figure 23b shows one such anomaly.

We detail the heuristics for computing the above six anomaly metrics, the thresholds and summary statistics of all anomalies found, in the supplementary section and the website. The anomalies found in the paper were cross-checked with the platform vendor Aerogram and the deployment partner, the public bus company DIMTS, for correctness and usefulness. All cases on inter-sensor and intra-sensor variations (metrics 5 and 6) were caused by local electrical maintenance work in a particular bus at the depot, whose sensor readings deviated from other buses in the depot. Lack of samples per minute or per hour (metrics 1 and 2) are helping to understand 4G networking issues. Finally the metrics for active hours per day and spatial coverage consistency (metrics 3 and 4) are helping to gain insights on unpredictable public bus behavior in Delhi, especially during Covid-19 induced lockdowns, where bus schedules and routes are seeing significant variations. Thus all these anomalies are highly important to gain insights about a live IoT network deployment. Defining these metrics and the multiple thresholds for them has been cumbersome, and more automated ML methods using this dataset and our findings as ground-truth, will be immensely valuable.

# H   Miscellaneous

We also observed the output of Spatio-temporal Interpolation using Random Forest (RF) algorithm to see the distribution of RMSE for the predictions. Fig. 24 shows some locations with large error, which needs indicates need of special handling.

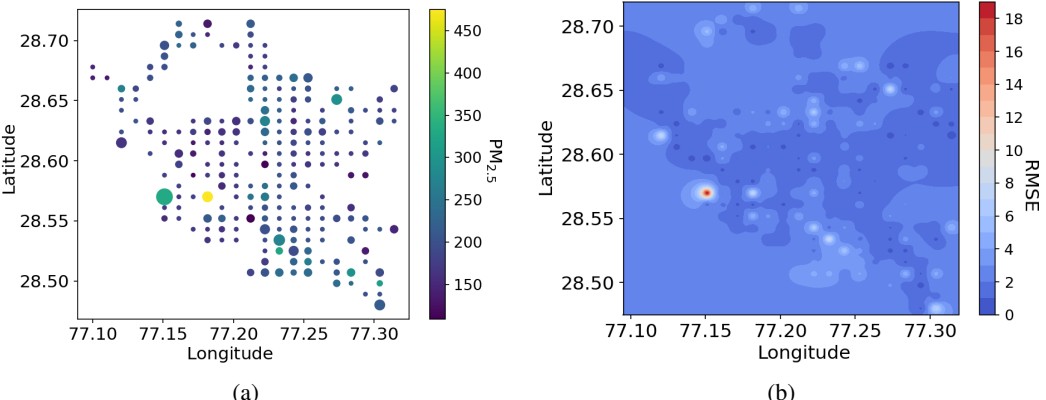

|  (a)  |  (b)  |

Figure 24: RMSE of the Spatio-temporal Interpolation using RF. (a) bigger circle denotes bigger RMSE for the spatial location while modeling. (b) shows the RMSE distribution. Few locations are hard to model.

# I Letters of Approval / Certifications from authorities

## I.1 ICAT EMC certification

ICAT EMC certification of our instrument verifying that it doesn't interfere with the bus's electro-mechanical properties.

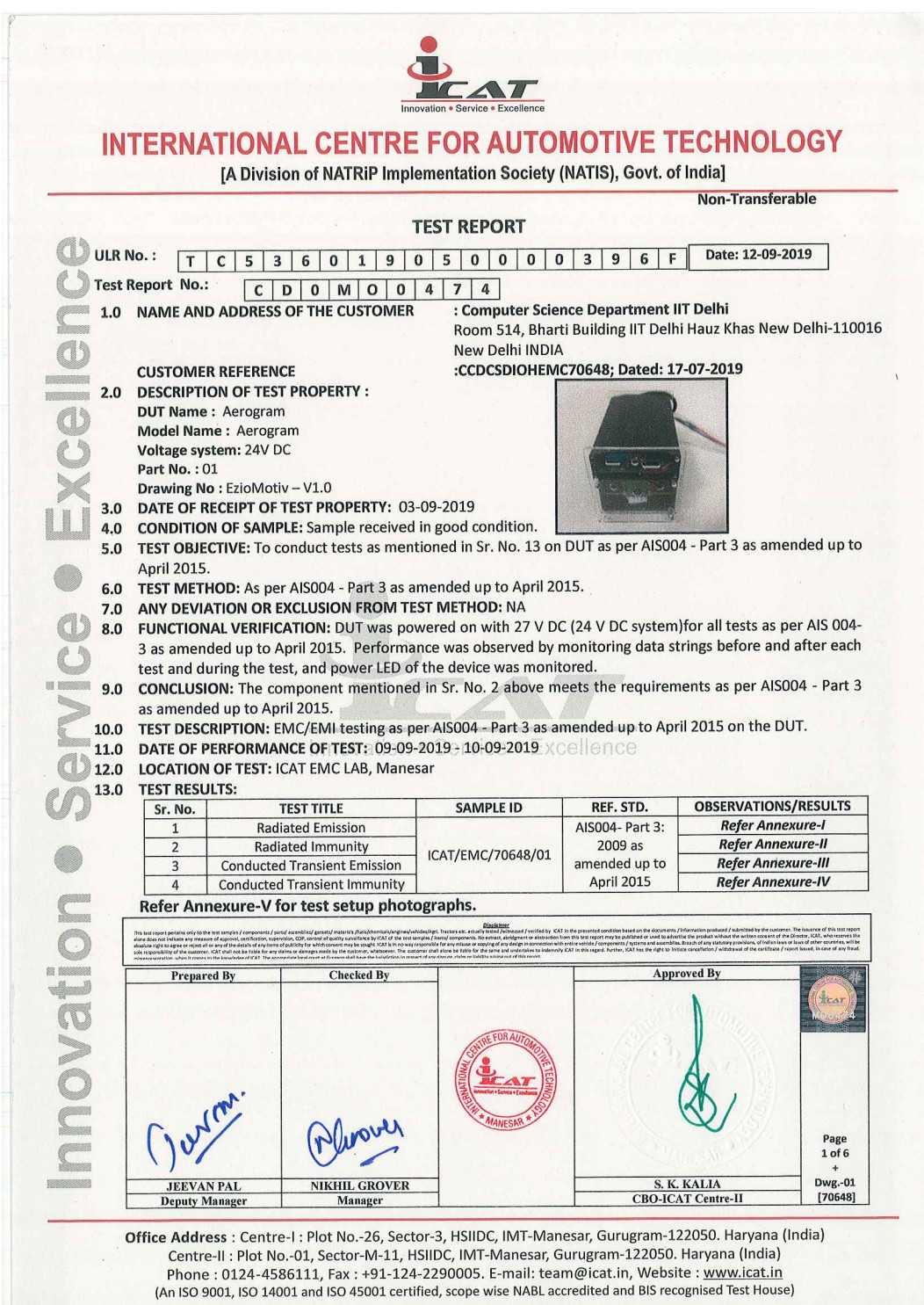

International Centre for Automotive Technology Test Report with details including ULR No., Test Report No., customer name and address, test property description, test results table, and signatures.

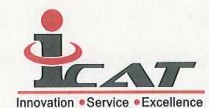

## 14.0 CLASSIFICATION OF FUNCTIONAL STATUS:

**CLASSIFICATION OF FUNCTIONAL STATUS AS PER (A.4) ANNEX-A, ISO 7637-2:2004:**

| CLASSES | DESCRIPTION |
|---|---|
| *CLASS A* | All functions of the device/system perform as designed during and after the test. |
| *CLASS B* | All functions of the device/system perform as designed during the test. However, one or more may go beyond the specified tolerance. All functions return automatically to within normal limits after the exposure is removed. |
| *CLASS C* | One or more functions of a device/system do not perform as designed during the test but return automatically to normal operation after the exposure is removed. |
| *CLASS D* | One or more functions of a device/system do not perform as designed during the exposure and do not return to normal operation until exposure is removed and the device/system is reset by simple 'operator/use' action. |
| *CLASS E* | One or more functions of a device/system do not perform as designed during and after exposure and cannot be returned to proper operation without repairing or replacing the device/system. |

## 15.0 LIST OF EQUIPMENTS USED IN THE TEST AND CALIBRATION DETAILS:

| Lab ID | Name of Instruments | Manufacturer | Model (S. No.) | Calib. due date |
|---|---|---|---|---|
| **Radiated Emission** | | | | |
| ICAT/EMC/TR - 01 | EMI Test Receiver | Rohde and Schwarz | ESU-8 (100290) | 03/05/2020 |
| ICAT/EMC/EPA-01 | External Preamplifiers | TDK RF Solutions | PA-02-001-100 (121054) | 03/05/2022 |
| ICAT/EMC/OFBA-04 | Biconical Antenna with polarization adaptor | TDK RF Solutions | PBA2030 (130818) | 09/05/2021 |
| ICAT/EMC/OFBA-05 | Broadband Log periodic Antenna | TDK RF Solutions | PLP 3003 (130830) | 09/05/2021 |
| ICAT/EMC/AN-01 | LISN | Schwarzbeck | NNBM8124 (8124-649) | 03/05/2022 |
| ICAT/EMC/AN-02 | | Schwarzbeck | NNBM8124 (8124-650) | 03/05/2022 |
| **Radiated Immunity** | | | | |
| ICAT/EMC/SG-01 | Signal Generator | Agilent Technologies | N5183A-520 (50140523) | 29/04/2022 |
| ICAT/EMC/SG-03 | | Agilent Technologies | SMB100A (103955) | 30/04/2022 |
| ICAT/EMC/AN-01 | LISN | Schwarzbeck | NNBM8124 (8124-649) | 03/05/2022 |
| ICAT/EMC/AN-02 | | Schwarzbeck | NNBM8124 (8124-650) | 03/05/2022 |
| ICAT/EMC/CIP-02 | Current injection probe | FCC | F -140 (130055) | - |
| ICAT/EMC/OFBA-07 | V Log Array Antenna | TDK RF Solutions | VLA-8001 (130835) | - |
| ICAT/EMC/OFBA-10 | Horn Antenna | TDK RF Solutions | ATH800M5GA (0337348) | - |
| ICAT/EMC/PM-01 | RF Power Meter | Agilent Technologies | N1914A (MY50000499) | 30/04/2022 |
| ICAT/EMC/PM-03 | | Agilent Technologies | N1912A (MY54010017) | 30/04/2022 |
| ICAT/EMC/AMP-01 | Amplifier | AR | 500W1000A (0335094) | - |
| ICAT/EMC/AMP-02 | | AR | 2500A225, Sr. No. 338773 | - |
| ICAT/EMC/AMP-03 | | AR | 500T1G2 (0336388) | - |
| ICAT/EMC/APS-01 | Average Power sensor (9kHz-6GHz) | Agilent Technologies | E9304 (S.No. MY51020021) | 30/04/2020 |
| ICAT/EMC/APS-02 | | Agilent Technologies | E9304 (S.No. MY51030004) | 30/04/2020 |
| ICAT/EMC/PS-01 | Power Sensor (50MHz-18GHz) | Agilent Technologies | N1921A (MY53380017) | 30/04/2020 |
| ICAT/EMC/PS-02 | | Agilent Technologies | N1921A (MY53380020) | 30/04/2020 |
| ICAT/EMC/FP-06 | Field Probe | AR | FL7018 (0334718) | 30/09/2020 |
| **Conducted Transient Emission** | | | | |
| ICAT/EMC/PG/05 | Voltage drop simulator | EM test | VDS 200N100 | 21/01/2020 |
| ICAT/EMC/DSO/01 | Digital Storage Oscilloscope | EM test | DSO9254A | 21/01/2020 |
| ICAT/EMC/AN/01 | Single line artificial network | EM test | AN 200N100 | 21/01/2020 |
| ICAT/EMC/SW/01 | Electronic switch | EM test | BS 200N100 | 21/01/2020 |
| ICAT/EMC/MR/01 | Matching resistor for transient generators | EM test | CAISO | 21/01/2020 |
| **Conducted Transient Immunity** | | | | |
| ICAT/EMC/PG/02 | Ultra-Compact Simulator | EM test | UCS 200N100 | 21/01/2020 |
| ICAT/EMC/PG/05 | Voltage drop simulator | EM test | VDS 200N100 | 21/01/2020 |
| ICAT/EMC/PG/05 | Load dump simulator | EM test | LD200N | 21/01/2020 |

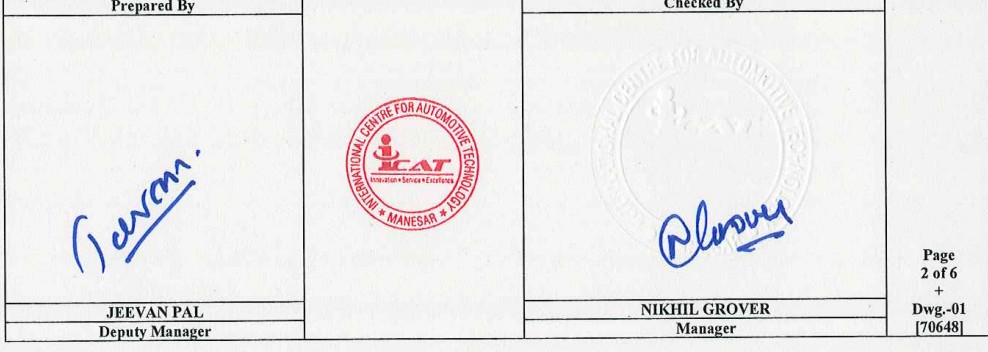

| Prepared By | Checked By | |
|---|---|---|
| | | Page 2 of 6 + Dwg.-01 [70648] |
| **JEEVAN PAL** Deputy Manager | **NIKHIL GROVER** Manager | |

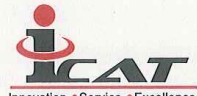

## Annexure – I

### 1.0 Measurement of Radiated Emissions:

#### 1.1 Test Condition:

| Operating Condition | Powered ON |
|---|---|

#### 1.2 Test Specifications:

| Frequency Range | 30MHz - 1000MHz |
|---|---|
| Step Size | 50kHz |
| Bandwidth | 120kHz |
| Measurement time | 5ms |
| Antenna | 30MHz - 300MHz: Bi-conical antenna, 300MHz - 1000MHz: Log-Periodic antenna |
| Antenna Polarization | Horizontal and Vertical |
| Antenna Location | In front of centre of harness |
| Antenna Distance | 1 meter |
| Detector | Peak and Average |
| Harness length | 1700mm |

#### 1.3 Test Graphs:

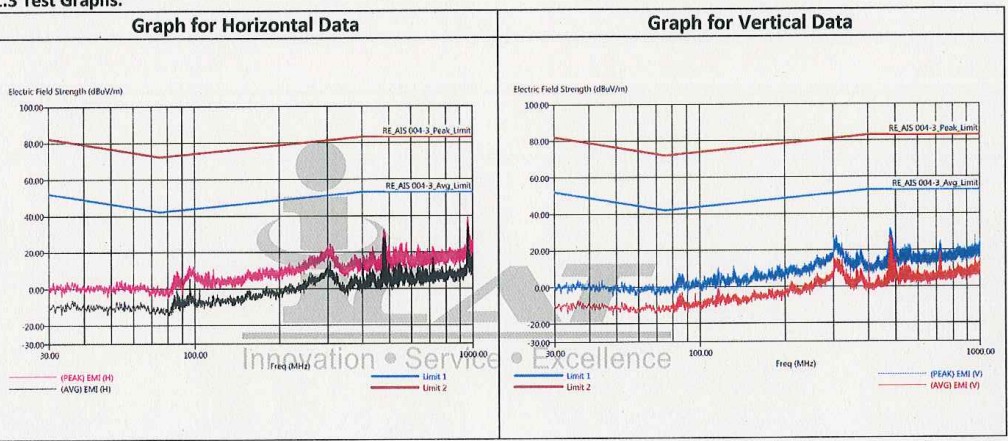

#### 1.4 Test Requirements:

Radiated Emissions measured should be within limits defined in AIS004- Part 3: 2009 as amended up to April 2015.

#### 1.5 Test Observations /Results:

Radiated Emissions measured are within limits.

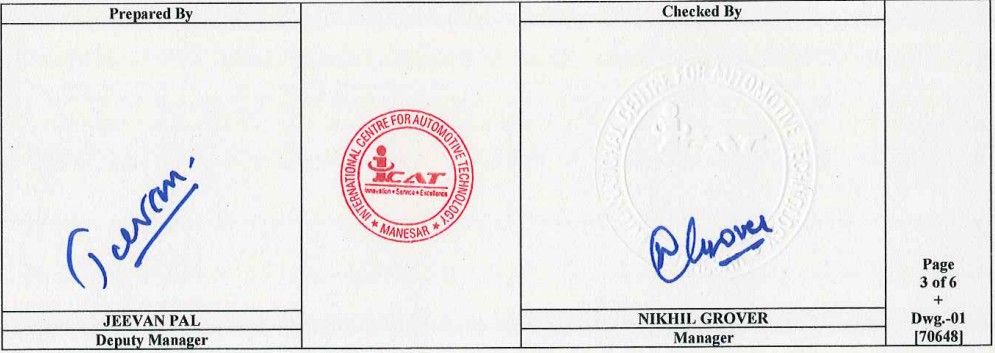

| Prepared By | | Checked By | |
|---|---|---|---|
| | | | Page 3 of 6 + Dwg.-01 [70648] |
| **JEEVAN PAL** Deputy Manager | | **NIKHIL GROVER** Manager | |

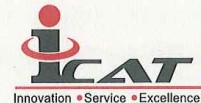

## Annexure – II

**2.0 Radiated Immunity Test:**

**2.1 Bulk Current Injection (BCI):**

**2.1.1 Test Condition:**

| Operating Mode | Powered ON |
|---|---|

**2.1.2 Test Specifications:**

| Frequency Range | 20MHz – 80MHz |
|---|---|
| Step Size | 5% |
| Current Severity Level | 60mA |
| Dwell Time | 2s |
| Harness length | 1700mm |
| Current probe position | 150mm from DUT |
| Test Method | Substitution (open loop) |
| Modulation | Amplitude modulation with 1 kHz modulating frequency and 80 % modulation depth. |

**2.1.3 Test Observations /Results:**

| S. No. | Frequency Range | Modulation | Acceptance Criteria | Observation/Result |
|---|---|---|---|---|
| 1. | 20MHz to 80MHz | Amplitude Modulation (AM) | No deviation in performance of DUT should be observed during test | No deviation observed |

**2.2 Absorber Lined Shielded Enclosure (ALSE) method:**

**2.2.1 Test Condition:**

| Operating Mode | Powered ON |
|---|---|

**2.2.2 Test Specifications:**

| Frequency Range | 80MHz – 2000MHz |
|---|---|
| Step Size | 80-400MHz: 5%, 400-2000MHz: 2% |
| Field Severity Level | 30V/m |
| Dwell Time | 2s |
| Harness length | 1700mm |
| Antenna | 80MHz – 1000MHz: V log array antenna, 1000MHz – 2000MHz: Horn antenna |
| Antenna Polarization | Vertical |
| Antenna Location | 80MHz – 1000MHz: in front of centre of harness, 1000MHz – 2000MHz: in front of DUT |
| Antenna Distance | 1 meter |
| Test Method | Substitution |
| Modulation | 80MHz – 800MHz: Amplitude modulation with 1 kHz modulating frequency and 80 % modulation depth
800MHz – 2000MHz: Pulse modulation: Ton: 577µs, period: 4600µs |

**2.2.3 Test Observations /Results:**

| Sr. No. | Frequency Range | Modulation | Antenna Polarization | Acceptance Criteria | Observation/Result |
|---|---|---|---|---|---|
| 1. | 80MHz to 800MHz | Amplitude Modulation | Vertical | No deviation in performance of DUT should be observed | No deviation observed |
| 2. | 800MHz to 1000MHz | Pulse Modulation | | | |
| 3. | 1000MHz to 2000MHz | Pulse Modulation | | | |

| Prepared By | Checked By | |
|---|---|---|
| | | |
| **JEEVAN PAL**
**Deputy Manager** | **NIKHIL GROVER**
**Manager** | Page 4 of 6 + Dwg.-01 [70648] |

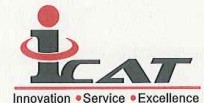

## Annexure – III

### 3.0 Measurement of Conducted Transient Emissions:

**3.1 Test Condition:**

| Operating Condition | Powered ON |
|---|---|

**3.2 Test Observations/Results:**
24V System

| Sr. No. | Supply Polarity | Limits as per AIS 004:Part 3 | Observation | Results |
|---|---|---|---|---|
| Fast transient | | | | |
| 1. | DUT ON to OFF | Positive: +150V
Negative: -400V | Positive Transient: No Significant Transient
Negative Transient: No SignificantTransient | Within Limits |
| 2. | DUT OFF to ON | | Positive Transient: 42.0 V
NegativeTransient: No SignificantTransient | Within Limits |
| Slow transient | | | | |
| 3. | DUT ON to OFF | Positive: +150V
Negative: -400V | Positive Transient: No Significant Transient
Negative Transient: No Significant Transient | Within Limits |
| 4. | DUT OFF to ON | | Positive Transient:43.32V
NegativeTransient: No SignificantTransient | Within Limits |

## Annexure – IV

### 4.0 Immunity to Transient Disturbances Conducted along Supply Lines as per AIS 004-3 as amended up to April 2015 following ISO 7637-2:2004:

**4.1 DUT Condition:**

| Operating Condition | Powered ON |
|---|---|

**4.2 Test Requirements and Observations/Results:**
24V System:

| Test Pulse | Severity Level | Acceptance Criteria | Achieved Class | Observations | Results |
|---|---|---|---|---|---|
| Pulse 1 | | Class C | Class C | Reset Observed during pulse injection | Satisfactory |
| Pulse 2a | | Class B | Class A | No deviation in performance observed | Satisfactory |
| Pulse 2b | III | Class C | Class C | Reset Observed during pulse injection | Satisfactory |
| Pulse 3a | | Class A | Class A | No deviation in performance observed | Satisfactory |
| Pulse 3b | | Class A | Class A | No deviation in performance observed | Satisfactory |
| Pulse 4 | | Class C | Class C | Reset Observed during pulse injection | Satisfactory |

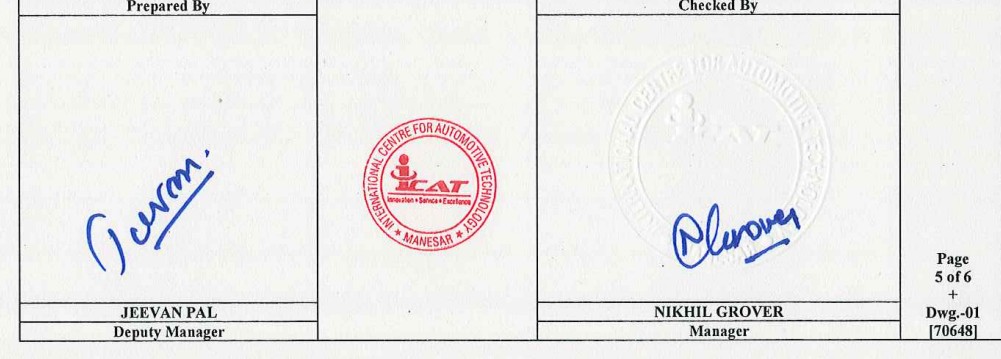

| Prepared By | | Checked By | |
|---|---|---|---|
| | | | |
| **JEEVAN PAL**
Deputy Manager | | **NIKHIL GROVER**
Manager | Page
5 of 6
+
Dwg.-01
[70648] |

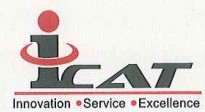

## Annexure – V

**5.0 Test Setup and Test Circuitry:**

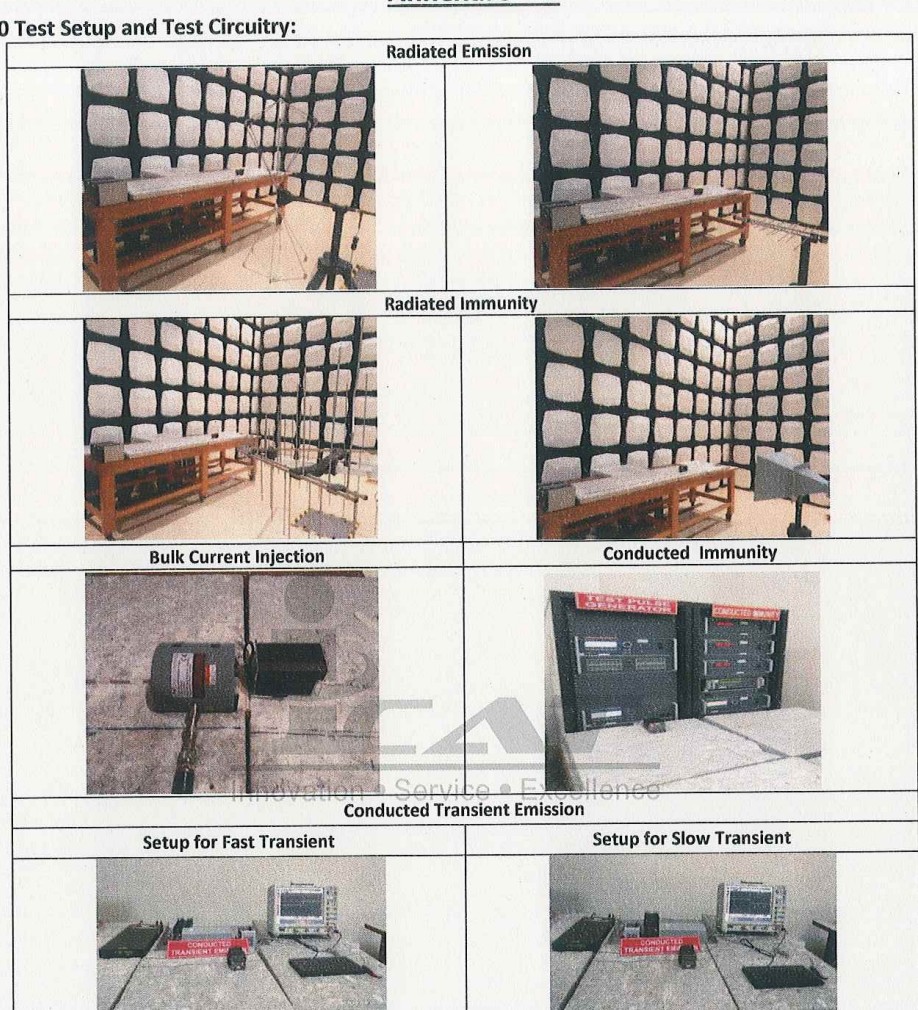

**Conducted Transient Emission**

| Setup for Fast Transient | Setup for Slow Transient |
|---|---|

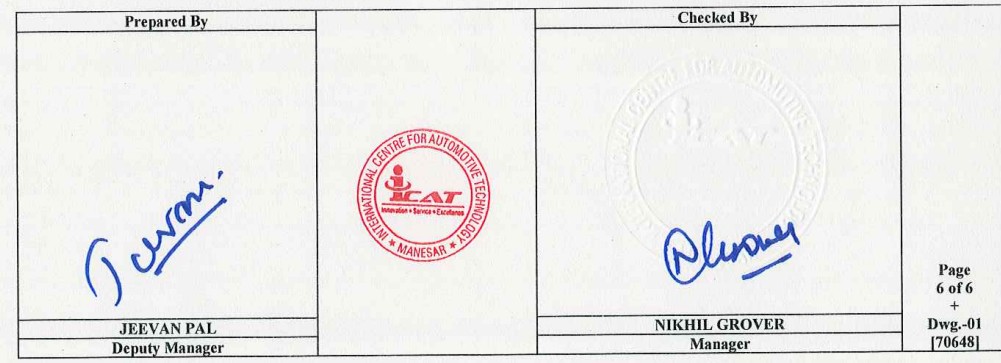

---------- **END OF REPORT** ----------

| Prepared By | | Checked By | |
|---|---|---|---|
| | | | |
| **JEEVAN PAL**
**Deputy Manager** | | **NIKHIL GROVER**
**Manager** | Page
6 of 6
+
Dwg.-01
[70648] |

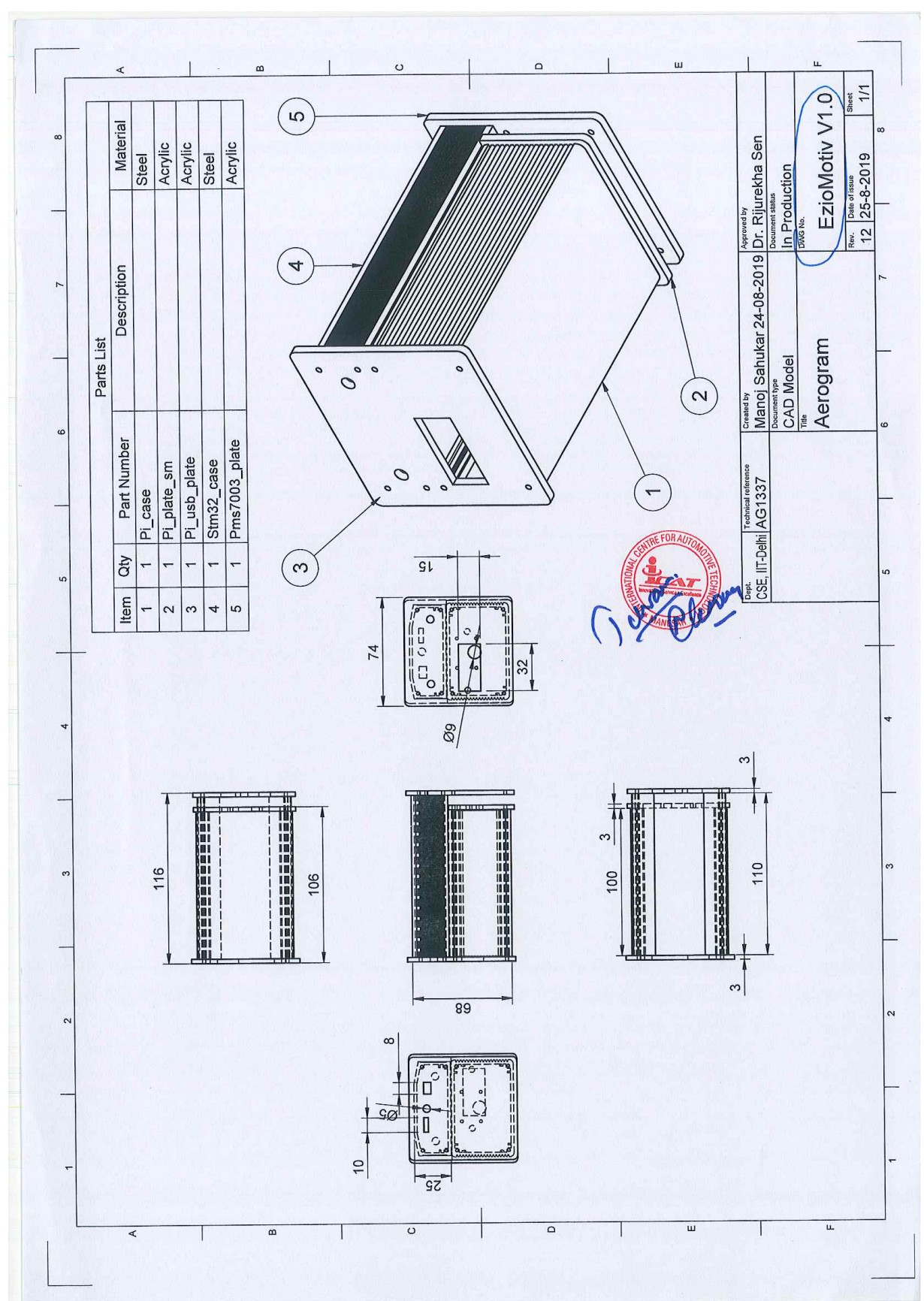

## Parts List

| Item | Qty | Part Number | Description | Material |
|------|-----|-------------|-------------|----------|
| 1 | 1 | Pi_case | | Steel |
| 2 | 1 | Pi_plate_sm | | Acrylic |
| 3 | 1 | Pi_usb_plate | | Acrylic |
| 4 | 1 | Stm32_case | | Steel |
| 5 | 1 | Pms7003_plate | | Acrylic |

| | | |
|---|---|---|
| Created by | Approved by | |
| Manoj Sahukar 24-08-2019 | Dr. Rijurekha Sen | |
| Document type | Document status | **EzioMotiv V1.0** |
| CAD Model | In Production | |
| Technical reference | Title | DWG No. |
| CSE, IIT-Delhi AG1337 | Aerogram | Rev. Date of issue Sheet |
| Dept. | | 12  25-8-2019  1/1 |

**I.2   Delhi Integrated Multi-Modal Transit System (DIMTS) letter of support**

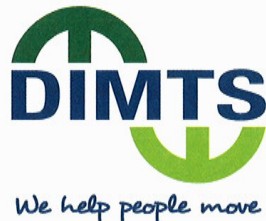

Ref: DIMTS/TP/2018/2756                                    Dated: June 21, 2018

To,
**Department of Science & Technology**
Delhi -

Subject :        **Letter of Support for the Proposed Research study.**

On behalf of DIMTS, we will extend our support to Profs Pravesh Biyani and Rijurekha Sen for their research proposal related to "Vehicle mounted Particulate Matter (PM) sensing in Delhi-NCR".

DIMTS runs more than 1600 non air-conditioned cluster buses on various routes in the Delhi region. We will facilitate the use of some of the vehicle fleet as needed by the researchers for pilot studies as they build and test their vehicle mounted sensing system.

Pollution being a pressing problem in Delhi-NCR, partnering with this research effort in a meaningful way will be very exciting for DIMTS.

Thanking you.

Yours faithfully,

**Samir Sharma**
Vice President - Transport Planning

**DELHI INTEGRATED MULTI-MODAL TRANSIT SYSTEM LTD.**
(A joint venture of the Govt. of NCT of Delhi and IDFC Foundation)
An ISO 9001:2015, ISO 14001:2015, OHSAS 18001:2007 & ISO 27001 certified and CMMI appraised company
CIN No. U60232DL2006PLC148406
REGD. OFF.: 1ST FLOOR, MAHARANA PRATAP ISBT BUILDING, KASHMERE GATE, DELHI 110 006 (INDIA)
Tel: +91 11 43090100 • Fax: +91 11 23860966 • Email : info@dimts.in • Web: www.dimts.in

**I.3 Delhi Pollution Control Committee (DPCC) letter of Support**

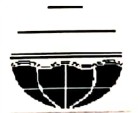

# Delhi Pollution Control Committee

5th Floor I.S.B.T. Building Complex Kashmere Gate Delhi 110006

Visit us at :http: //dpcc.delhigovt.nic.in

F. No. DPCC|(12)(1)(260)Lab(A) 2020|2203           Date: 27/1/2020

To,

Dr. Rijurekha Sen
Department of Computer Science,
IIT Delhi, Hauz Khaz,
New Delhi-110016

Subject- Support Letter for Vehicle Mounted Low Cost PM Monitoring in Delhi

Madam,

With reference to your E-mail and telephonic discussion this organization is interested to know feasibility of Vehicle Mounted Low Cost PM Monitoring in Delhi and willing to share data generated by DPCC Ambient Air Quality Network to assess error percentage of Low Cost System.

(Dr. M. P. George)
Scientist
Dr. M. P. GEORGE
Scientist

**I.4    Delhi Ministry of Transport (MOT) Permission**

**GOVERNMENT OF NCT OF DELHI**
**TRANSPORT DEPARTMENT**
**(CLUSTER & DTC SECRETARIAT)**
**5/9, UNDER HILL ROAD, DELHI – 110 054**

No. F.10/STA/Policy /Tpt./ 2011/ 333/40631                      Date: 17/08/2020

To

The CEO,
Delhi Integrated Multi Modal Transit System Ltd.,
8th Floor, Block-1, Delhi Technology Park,
Shastri Park, Delhi-110053.

**Subject: Request for permission to install pollution sensing units in Cluster buses as a part of R&D Project by IIT, Delhi.**

Sir,

Kindly refer to your letter no. DIMTS/Road Transport/2019/4398, dated 05.11.2019, on the abovementioned subject. DIMTS had requested for a formal approval to install pollution sensing units in 10 Cluster buses by CSE IIT, Delhi.

In this context, I am directed to convey the approval of Hon'ble Minister (Transport) for installing of pollution sensing units in 10 Cluster buses of the Kushak Nalah Depot by CSE, IIT Delhi.

Yours faithfully,

**(Subodh Kumar)**
**Deputy Commissioner**
**(Cluster & DTC Sectt.)**

**Copy to:**
1. Dy. Commissioner (PCD) with reference to U.O. NO. 23(1471)/CAP/TPT/PCD/ 2018/ 1595/87542 dated 26.11.2019.
2. Ms. Rijurekha Sen, Assistant Professor, CSE, IIT, Delhi.
3. M/s. Indraprastha Logistics Pvt. Ltd, 80/2, Ground Floor Govindpuri Kalkaji New Delhi-110019

## I.5   Letter of funding: SCIENCE & ENGINEERING RESEARCH BOARD (SERB), INDIA

FILE NO. IMP/2018/001481
### SCIENCE & ENGINEERING RESEARCH BOARD (SERB)
(A statutory body of the Department of Science & Technology, Government of India)

5 & 5A, Lower Ground Floor
Vasant Square Mall
Plot No. A, Community Centre
Sector-B, Pocket-5, Vasant Kunj
New Delhi-110070

Dated: 29-Mar-2019

## ORDER

Domain: Information & Comm. Technology
Subject: Financial Sanction of the research project titled "Scalable Spatio-Temporal Measurement and Analysis of Air Pollution Data for Delhi-NCR using Vehicle-Mounted Sensors " under the guidance of Dr. Rijurekha Sen, Department of Computer Science, Indian Institute of Technology Delhi , Hauz Khas, New Delhi, DELHI-110016 and by Dr. Pravesh Biyani, Assistant Professor, Ece Dept, Indraprastha Institute Of Information Technology and by Dr. Arnab Bhattacharya, Associate Professor, Department Of Computer Science And Engineering, Indian Institute Of Technology Kanpur and by Dr. Sayan Ranu, Assistant Professor, Computer Science And Engineering, Indian Institute Of Technology Delhi  - Release of 1st grant.

Sanction of **Science and Engineering Research Board (SERB)** is hereby accorded to the above mentioned project at a total cost of **Rs. 12746800/- (Rs. One Crore Twenty Seven Lakh Forty Six Thousand Eight Hundred Only)** with break-up of **Rs. 5500000/-** under **Capital (Non-recurring)** head and **Rs. 7246800/-** under **General (Recurring)** head for a duration of 36 months. The items of expenditure for which the total allocation of **Rs. 12746800/-** has been approved are given below:

| S. No | Head | Total (in Rs.) |
|---|---|---|
| A | Non-recurring | |
| 1 | Equipment
-> Laptop
-> Server
-> Sensors | 5500000 |
| A' | Total (Non-Recurring) | 5500000 |
| B | Recurring Items | |
| 1 | Recurring - I : (Manpower)
Recurring - II : ( Consumables, Travel, Contingencies)
Recurring - III : Scientific Social Responsibility | 3888000
2200000
0 |
| 2 | Recurring - IV : (Overhead Charges) | 1158800 |
| B' | Total (Recurring) | 7246800 |
| C | Total cost of the project (A' + B') | 12746800 |

2.  Sanction of the SERB is also accorded to the payment of  **Rs. 5500000/- (Rupees Fifty Five Lakh only)** under 'Grants for creation of capital assets' and **Rs. 2415000/- (Rupees Twenty Four Lakh Fifteen Thousand only)** under 'Grants-in-aid General' to IRD, **Indian Institute Of Technology Delhi, Hauz Khas, New Delhi**  being the first installment of the grant for the year 2018-2019 for implementation of the said research project.

3. The expenditure involved is debitable to  **Fund for Science & Engineering Research (FSER)**
**This release is being made under Impacting Research Innovation and Technology (IMPRINT-2). (PAC Information & Communication Technology)**

4. The Sanction has been issued to Indian Institute Of Technology Delhi, Hauz Khas, New Delhi with the approval of the competent authority under delegated powers on **28 March, 2019** and vide Diary No. **SERB/F/13078/2018-2019** dated **28 March, 2019**

5. Sanction of the grant is subject to the conditions as detailed in Terms & Conditions available at website ( www.serb.gov.in).

6. Overhead expenses are meant for the host Institute towards the cost for providing infrastructural facilities and general administrative support etc. including benefits to the staff employed in the project.

7. While providing operational flexibility among various subheads under head Recurring-II, It should be ensured that not more than Rs. 450000 under Travel and Rs. 450000 under Contingency should be spent.

8. As per rule 211 of GFR, the accounts of project shall be open to inspection by sanctioning authority/audit whenever the institute is called upon to do so.

9. The sanctioned equipment would be procured as per GFR and its disposal of the same would be done with prior approval of SERB.

10. The release amount of **Rs. 7915000/- (Rupees Seventy Nine Lakh Fifteen Thousand only)** will be drawn by the Under Secretary of the SERB and will be disbursed by means of RTGS transaction as per their Bank details given below:

| | |
|---|---|
| **Account Name** | IRD ACCOUNTS IITD |
| **Account Number** | 10773572600 |
| **Bank Name & Branch** | STATE BANK OF INDIA IIT BRANCH, IIT HAUZ KHAS, NEW DELHI - 110016 |
| **IFSC/RTGS Code** | SBIN0001077 |
| **Email id of A/C Holder** | arird@admin.iitd.ac.in |
| **Email id of PI** | riju@cse.iitd.ac.in |

11.The institute will furnish to the SERB separate Utilization certificate(UCs) financial year wise to the SERB for Recurring (Grants-in-aid General) & Non-Recurring (Grants for creation of capital assets) and an audited statement of

accounts pertaining to the grant immediately after the end of each financial year.

12. The institute will maintain separate audited accounts for the project. A part or whole of the grant must be kept in an interest earning bank account which is to be reported to SERB. The interest thus earned will be treated as credit to the institute to be adjusted towards further installment of the grant.

13. The project File no. IMP/2018/001481 should be mentioned in all research communications arising from the above project with due acknowledgement of SERB.

14. The manpower sanctioned in the project, if any is co-terminus with the duration of the project and SERB will have no liability to meet the fellowship and salary of supporting staff if any. beyond the duration of the project

15. As this is the first grant being released for the project, no previous U/C is required.

16. The institute may refund any unspent balance to SERB by means of a Demand Draft favoring "FUND FOR SCIENCE AND ENGINEERING RESEARCH" payable at New Delhi.

17. The organization/institute/university should ensure that the technical support/financial assistance provided to them by the Science & Engineering Research Board should invariably be highlighted/ acknowledged in their media releases as well as in bold letters in the opening paragraphs of their Annual Report.

18. In addition, the investigator/host institute must also acknowledge the support provided to them in all publications, patents and any other output emanating out of the project/program funded by the Science & Engineering Research Board.

*Monika Agarwal*

(Dr. Monika Agarwal)
Scientist E
ms.imprint@gmail.com

To,
**Under Secretary**
**SERB, New Delhi**
**Copy forwarded for information and necessary action to: -**

| | |
|---|---|
| 1. | The Principal Director of Audit, A.G.C.R.Building, IIIrd Floor I.P. Estate, Delhi-110002 |
| 2. | Sanction Folder, SERB , New Delhi. |
| 3. | File Copy |
| 4. | Dr. Rijurekha Sen
Department of Computer Science
Indian Institute of Technology Delhi , Hauz Khas, New Delhi, DELHI-110016
Email: riju@cse.iitd.ac.in
Mobile: 919810591052

Dr. Pravesh Biyani
Ece Dept
Indraprastha Institute Of Information Technology

Dr. Arnab Bhattacharya
Department Of Computer Science And Engineering
Indian Institute Of Technology Kanpur

Dr. Sayan Ranu
Computer Science And Engineering
Indian Institute Of Technology Delhi
(Start date of the project may be intimated by name to the undersigned. For guidance, terms & Conditions etc. Please visit www.serb.gov.in.) |
| 5. | IRD,
Indian Institute Of Technology Delhi, Hauz Khas, New Delhi
(Receipt of Grant may be intimated by name to the undersigned) |
| 6. | Secretary,
Department of Science & Technology
Ministry of Science and Technology
Technology Bhavan, New Mehrauli Road,
New Delhi-110016
Email: dstsec@nic.in |
| 7. | Secretary (Higher Education)
Ministry of Human Resource Development
Shastri Bhavan, New Delhi- 110 001
Email: secy.dhe@nic.in |

*Monika Agarwal*

(Dr. Monika Agarwal)
Scientist E
ms.imprint@gmail.com