# OpenReview forum: "AirDelhi: Fine-Grained Spatio-Temporal Particulate Matter Dataset From Delhi For ML based Modeling"
_NeurIPS.cc/2023/Track/Datasets_and_Benchmarks — NeurIPS 2023 Datasets and Benchmarks Poster_

### Official Review · Reviewer_8Gda · 2023-07-03
**Overall, a valuable dataset - but related work and discussion needs work**

**Rating:** 6
**Confidence:** 4
**Correctness:** see strengths above
**Clarity:** see strengths above

**Strengths:**

* The findings of how the dataset varies from previous dynamic sensing networks is valuable for future work studying pollution in highly-polluted cities like Delhi
* ML interpolation and forecasting benchmarks shows that traditional ML methods are sufficient to learn the patterns in the data


**Additional Feedback:**

see weaknesses

**Documentation:**

The website for the data is sufficient for reproducbility.

**Ethics:**

None.

**Limitations:**

The variance in the data due to noisy low-cost sensors and the problem of calibration needs to be discussed. Further, the impact of the position of the sensor box inside the bus, as opposed to ambient pollution sensors that are not within a vehicle needs to be analyzed.

**Opportunities For Improvement:**

* The authors should discuss how this work is motivated or related to [1, 3] to avoid violations of the double submission policy between Neurips and ACM COMPASS/Mobicom.

* Further, the difference between static [2] and dynamic pollution maps of Delhi needs to be expanded in the discussion.

[1] Complexity of Factor Analysis for Particulate Matter (PM) Data: A Measurement Based Case Study in Delhi-NCR. Ismi Abidi,
Sagar Ravi Gaddam, Saswat Kumar Pujari, Chinmay Shirish Degwekar, Rijurekha Sen. ACM COMPASS '22

[2] Modeling fine-grained spatio-temporal pollution maps with low-cost sensors. Shiva R. Iyer, Ananth Balashankar, William H. Aeberhard, Sujoy Bhattacharyya, Giuditta Rusconi, Lejo Jose, Nita Soans, Anant Sudarshan, Rohini Pande & Lakshminarayanan Subramanian. NPJ Climate Change and Sustainability '22.

[3] Low Cost Platform Design for Pollution Measurement in Delhi-NCR using Vehicle-Mounted Sensors. Tanishka Goyal, Ankita Singh,
Smriti Chhaya, Aditi Vikas, Poorva Garg, Ritika Malik, Rijurekha Sen. Mobicom '18



**Relation To Prior Work:**

see opportunities for improvement, with special attention to how this work differs from specific papers in 2018 and 2022 from some of the same authors on the topic

**Summary And Contributions:**

* The paper provides a valuable dataset of particulate matter gathered from low-cost sensors mounted on public buses in Delhi.

---

> ### Author Response · Authors · 2023-08-25
> **Response to Reviewer 8Gda**
>
> We thank the reviewer for critical review of our work and providing us an opportunity for further improvements through constructive feedback. We also appreciate for recognizing our work for *being different from previous dynamic sensing networks in highly polluted cities like Delhi which is valuable for future work studying pollution, with ML interpolation and forecasting benchmarks showing that traditional ML methods are sufficient to learn the patterns in the data*.
>
> We have updated the main manuscript and the appendix to address these comments. The changes made are highlighted in blue colour. A comprehensive response to your feedback is presented below.
>
> **(Q1) Discuss work relation to [1, 3] for double submission policy between Neurips and ACM COMPASS/Mobicom**
>
> The [3] presents the complete design of the low-cost platform. We have updated proper reference in section 3.1 (P3:L122). We later performed rigorous tests for automotive safety certification and received appropriate permissions and letters of support from the Delhi Ministry of Transport and Delhi Pollution Control Committee, which are provided with the *Appendix I*. We mounted pollution tracking sensors on the permissible 13 public buses in Delhi for 3 months (Nov 1, 2020 to Jan 31, 2021), in collaboration with Delhi Integrated Multimodal Transport System and gathered the PM data. The [1] used similar low-cost sensors to analyze the relation of PM with static (green cover, buildup, commercial, residential) and dynamic (meteorological, traffic) factors, particularly at traffic intersections with odd-even policy in Delhi. We have discussed this in *Section 2 (P3:L105-107)*.
>
> Also, the [2] uses low-cost static sensors in Delhi to complement the CPCB sensors to train on longer duration data and predict on different longer timeline. In contrast, we use low-cost sensors in mobile deployment, and utilize running prediction using recent data to predict for today or next-day. We have updated the *Section 2 - Related Work (P3:L101-102)* referencing this work.
>
> **(Q2) Difference between static [2] and dynamic pollution maps**
>
> Static sensors are better in data quality, endurance and temporal resolution, a low-cost static sensor is better in temporal resolution which can provide (static) pollution maps with better temporal resolution for the installed locations. But the effective static pollution maps require careful placement of sensor nodes in large quantity leading to resource wastage.
>
> Mobile sensors are better in mobility, geographic coverage, maintenance, cost-efficiency which can provide (dynamic) pollution maps with greater geographic coverage, with limitation of communication overheads and redundant sampling. We have added the discussion in *Section 2 Related Work (P3:L107-113)*.
>
>
> **[Limitations] Data variance, calibration, positioning of sensor box**
>
> In *Section 3.2*, we compare the data collected with reference sensors, showing the accuracy and reliability in detail. In the data quality analysis in *Fig 2-3 (P4-5)* and through Anova statistical test in *Appendix C*, we show that our low-cost sensors are performing similar to high grade Dustrak and static sensors. We understand and appreciate the reviewer’s concern that these sensors may need early calibrations as compared to reference sensors. But the colocation experiments presented in *Fig 2* were performed in July 2021 after deployment and data collection was over, and the sensors were brought to the lab. We have analyzed them to be working sound after ~6 months in the polluted field. So, in our understanding, the presented dataset has minimal (if not none) impact, by the variance or calibration concerns.
>
> Also, these sensors are sensing the air the actual commuters breathe. Their readings, if any different due to the placement, should be more realistic and accurate than sensors placed at heights from the ground and traffic intersections. We have also updated appropriate discussion on Limitations in *Section 5 (P10:L351-358)*.
>
> *We thank you again for your time to understand our work and provide constructive insights. We trust that these revisions will satisfactorily address the concerns raised and elevate the overall quality of our work to an acceptable standard.*

---

### Official Review · Reviewer_Q5NU · 2023-07-20
**A well-motivated and novel dataset**

**Rating:** 8
**Confidence:** 4

**Strengths:**

The paper is very well-motivated and it addresses a pressing real-world challenge. For example, in Intro, it is stated that 50% of all children in the Delhi region suffer from irreversible lung damage.

The paper also very well discusses its main implications. For example, it states in the Conclusion that the aim is to recommend suitable locations for installing new expensive stationary sensors. Also, the method could be used to recommend to people which commuting/travel routes to take and when to wear masks.

The contributions are mainly well-stated.

The frequency at which samples are collected.

**Additional Feedback:**

I encourage the authors to keep up working on this challenging but tremendously important task not only for public health in Delhi and India, but the results of which could be translated to other countries, as well.

**Clarity:**

Yes, the paper is well written and presented, except for a couple of minor points raised under opportunities for improvement.

**Correctness:**

The benchmarks are many and they mostly seem to be conducted soundly. One issue is that section 4.1 is not easy to follow. Specific issues are discussed below.

Section 4.1 Dataset Pre-processing and Evaluation Metrics for the Analysis is not clear. In particular, how the data were split into K-Folds, and why (an explanation for this seems to come after and it is still not completely clear. The explanation should occur in this section before K-Folds are introduced). I appreciate the authors introducing Figure 4 to visualise the splits but Figure (a) is not clear.

Figure 4: PM Data Splits. -- It is not clear if this is an example of a possible split. Why the cube edges do not match from spatial to temporal? The caption could be longer to elaborate more on this complicated Figure.

**Documentation:**

The dataset-creation is mainly well documented in this work. It is also discussed in a previously published paper by the same authors that is referenced.

**Ethics:**

The authors have received many permission documents from the Delhi authorities, which are provided in the Supplementary Information. Possible beneficial uses of the dataset are mentioned in Discussion. Except for stigmatising certain parts of the city, which was not mentioned, the reviewer does not see any other problematic uses of the dataset that need to be discussed.

**Limitations:**

While this work does not raise explicit potential negative societal implications, there are several limitations that deserve to be discussed.

Perhaps it warrants a mention that the dataset is collected during COVID time under limitations.


Some discussion on the uneven spatial distribution of the dataset (i.e., which is constrained by the bus routes) should be included, e.g., under limitations.

The dataset covers a period of 3 months, which is another potential limitation to mention.

**Opportunities For Improvement:**

The conclusion in the abstract that this dataset differs from the existing datasets weakens the contribution without mentioning that this dataset's quality was also validated against existing expensive sensors. This should be mentioned in the abstract.


Section 4.2: sometimes uses PM2.5 and sometimes PM. It would be better not to mix them, but choose one notation and stay consistent if possible.

The paper does not explain or at least speculate why the Delhi dataset could be so different when it comes to some of the benchmarking tasks compared to similar datasets from other countries. It could be that the levels of pollution in Delhi are much more varied, in addition to being higher, which causes this. Or it could be the fact that the data are sampled at random locations (that are affected by the varying speeds at which buses take their routes). But this is not obvious and is worth discussing in the paper and perhaps under limitations if the authors so choose.

I am not convinced that the Related Work should be discussing other methods only, instead, it should mainly discuss other similar datasets. I assume the methods currently found in Related Work can also be covered but since the main contribution of the paper is the datasets, related work should also discuss the related datasets (such as the one used in benchmarking later).

A suggestion to the authors -- Figure 2 b) indeed shows that the differences are negligible; but perhaps it would be better visualised if the x-axis had the range as the y-axis on Figure 2 a). Or there could be another way to confirm this claim, for example by expressing the median/average value of the mean differences in percentage points of the mean PM value across the whole dataset.


Figure 3: There could be more explanation about the heatmap. It is a heatmap of which period? Of one bus route or more? Or is it for a single day (I guess this is from the paragraph Heat Map, but that information should be also found in the Caption)?



At the beginning of Section 4.1 these tasks/benchmarks should be first defined: Spatio-temporal Interpolation, Spatio-temporal Missing Data Imputation, and Spatio-temporal Forecasting so that it is clear how they are operationalized. In particular, it is not clear how Interpolation and Missing Data Imputation differ in this context.

**Relation To Prior Work:**

The paper references related benchmark methods well. Other datasets similar to the one contributed in this paper are also discussed in Section 3.3. but they should be introduced in Related Work first.

**Summary And Contributions:**

In this dataset and benchmark paper, an Air Quality (for various PM indicators) dataset is collected for Delhi. The authors built and benchmarked their own low-cost sensors, that were then deployed on local buses. Here, 3 months collection is presented and used as a benchmark for 3 types of tasks: Spatio-temporal Interpolation, Spatio-temporal Missing Data Imputation, and Spatio-temporal Forecasting.

The main contributions are 1) the novel dataset, validated against existing expensive and static sensors from the city authorities; 2) the benchmarking results showing that this dataset is more "challenging" compared to similar datasets from other countries and 3) it can be used for ML modelling requiring novel approaches potentially.

---

> ### Author Response · Authors · 2023-08-25
> **Part 1/2 for Reviewer Q5NU**
>
> We thank the reviewer for critical review of our work and providing us an opportunity for further improvements through constructive feedback. We also appreciate for recognizing our work for *being a well-motivated paper to addresses a pressing real-world challenge, with well-stated contributions and data collection frequency and having good implications*.
>
> We have updated the main manuscript and the appendix to address these comments. The changes are highlighted in blue colour. A comprehensive response to your feedback is presented below.
>
> **(Q1) Improve the abstract**
>
> We appreciate the suggestion, and updated the abstract for the dataset’s quality been validated against existing expensive sensors *(P1:L15-16)*.
>
> **(Q2) Consistent with PM notation**
>
> We updated PM to PM2.5 for consistency.
>
> **(Q3) Why the Delhi dataset different for benchmarking tasks compared to similar datasets from other countries**
>
> One thing is evident that the levels of pollution in Delhi are high and more varied. The added correlation and covariance discussion in *Appendix D* shows different pollution traits in Delhi, especially the two types of PM patterns. The meteorological analysis in *Appendix E* shows the impact of such meteorological factors and additionally points to some episodes explained by external non-meteorological factors, like stubble burning. The different bus-route analysis in *Appendix F* shows some routes with high pollution concentration than others.
>
> We also observe that the modeling errors increase with increasing PM2.5 levels. These make the Delhi dataset complex to model and unique in characteristics. We have discussed this in detail with figure depiction *(Fig 6) in Section 4.4 in Observation 1 (P8:L302-308)*.
> We have also updated appropriate discussion on Limitations in *Section 5 (P10:L350-356)*.
>
> **(Q4) Related Work should should also discuss the related datasets (like used in benchmarking later)**
>
> We have improved the *Section 2 Related Work* for the datasets and other concerns *(P3:L97-98,L102-113)*.
>
> **(Q5) Figure 2b indeed shows that the differences are negligible, do range of x-axis of 2b = y-axis of 2a. Or expressing the median/average value of the mean differences in percentage points of the mean PM value across the whole dataset**
>
> We have added plots in *Fig 2 in Section 3.2 (P4)* for comparison in PM2.5 levels between standard TSI Dustrak and our low-cost sensor. The *Fig 2b* shows the difference of hourly mean PM2.5 between DustTrak and one mobile sensor, the mean difference is 6.16%.
>
> *Fig 2c* shows the histograms of hourly mean PM2.5 for the shown PM2.5 intervals for the 3 devices, having x-axis range as the y-axis on Figure 2a (as suggested).
>
> Also, *Fig 9* in Appendix C also shows similar mean and standard deviation between the 3 devices supporting the Fig 2.
>
> **(Q6) Figure 3: More explanation about heatmap, which period, of one bus route or more, a single day or more**
>
> The Heat Map shown in *Fig 3(b) (P5)* is for all bus routes for Dec 15, 2020. We have improved the caption accordingly similar to the information available in the description in the related paragraph.
>
> **(Q7) At the beginning of Section 4.1 these tasks/benchmarks should be first defined: Spatio-temporal Interpolation, Spatio-temporal Missing Data Imputation, and Spatio-temporal Forecasting so that it is clear how they are operationalized. In particular, it is not clear how Interpolation and Missing Data Imputation differ in this context**
>
> As per your suggestions, we reordered the *Sections 4.1, 4.2, 4.3* with some updates to express the data preprocessing effectively. We hope it would suit other reviewers also.
>
> In *Section 4.1 (b)*, we can note that Missing Data Imputation also uses all locations data for T training days. The *(a)* spatio-temporal Interpolation only uses the visible set of locations. So, in missing data imputation only 1 day data from  T+1 th day is missing, but in interpolation, data from all of the T+1 days can be missing.

---

> ### Author Response · Authors · 2023-08-25
> **Part 2/2 for Reviewer Q5NU**
>
> **[Limitations]**
>
> **(L1) a mention that the dataset is collected during COVID time under limitations**
>
> In Delhi, the COVID restrictions were started to ease during the time the dataset was collected, limiting high restrictions to containment zones at local level only. Due to this, the impact of the COVID’19 on the collected dataset is expected to be minimal, if not none. We have added this in *Section 3.1 (P3:L132-134)*.
>
> **(L2) mention uneven spatial distribution (constrained by the bus routes) under limitations**
>
> The bus routes selected by us covered the different types of areas, including green cover, residential, commercial, termed as static factors and we have tried our best to reduce the bias towards any particular factor. We acknowledge that the data is though limited to the region from routes of the bus as they go through arterial roads, without deeper penetration in residential areas. An alternative could have been to put sensors in private vehicles, but it is not reasonable to drive cars just to gather the data. Also, commercial cabs can be used for data collection, which we can check in our future work. We have added the same in *Section 5 Limitations and Future Work (P10:L351-358)* and also added bus-routes analysis in *Appendix F*.
>
> **(L3) dataset period of 3 months, another potential limitation to mention**
>
> We acknowledge the limitation of 3 months duration, and in our next project in another metropolitan city in India, we are working for a comprehensive 6+ months data collection. Also, despite being a 3 month dataset, we are anyhow moving from a high limitation of few data points in the same vicinity to million+ points. Our dataset contains 12+ million PM2.5 samples, where existing datasets have less than 50K samples, making our dataset largest in its kind. We have added the same discussion under limitations in *Section 5 (P10:L351-358)*.
>
> **[Correctness] section 4.1 is not easy to follow**
>
> As discussed earlier, we have reordered the *Sections 4.1, 4.2 and 4.3* to improve the structure and readability. We have added/updated text in these sections to improve it further. We also updated the *Fig 4(a) (P8)* and its caption to clarify the different data splits of A,B,C,P.
>
> **[Relation To Prior Work] Other similar datasets discussed in Section 3.3. should be introduced in Related Work first**
>
> As answered in Q4, we have improved the *Section 2 Related Work* for the datasets and other concerns *(P3:L97-98,L102-113)*.
>
> **[Ethics] Except for stigmatising certain parts of the city, which was not mentioned, the reviewer does not see any other problematic uses of the dataset that need to be discussed**
>
> We have added related discussion in *Section 5 Limitations and Future Work (P10:L351-358)* and also added bus-routes analysis in *Appendix F* which exhibits the high PM levels for some routes. We hope that our dataset and analysis bring the attention to improve the pollution situation in the affected areas.
>
> *We thank you again for your time to understand our work and provide constructive insights. We trust that these revisions will satisfactorily address the concerns raised and further elevate the overall quality of our work to an acceptable standard*.

---

> > ### Comment · Reviewer_Q5NU · 2023-08-29
> >
> > Thanks to the reviewers for addressing the comments. Since they addressed all my comments, and this is one of the most valuable datasets that can be provided nowadays, given raising concerns about the influence of air pollution on public health, I will raise my score accordingly.

---

> > > ### Author Response · Authors · 2023-08-29
> > > **Thank you**
> > >
> > > We are happy to note the increase in score. We thank the reviewer for the insightful comments on our work, which has further elevated the quality of the manuscript.
> > >
> > > regards,
> > >
> > > Authors

---

### Official Review · Reviewer_npPd · 2023-07-22
**Contributions to air pollution tracking in emerging economies**

**Rating:** 6
**Confidence:** 3
**Correctness:** See Limitations
**Clarity:** Yes.

**Strengths:**


- Provision of publicly accessible PM2.5 and PM10 dataset for air pollution analysis
- Provision of benchmarking exercise, evaluating state-of-the-art methods for interpolation, feature imputation, and forecasting on this dataset

**Additional Feedback:**

- The authors briefly mentioned the difficulties of packaging the sensor but did not provide detailed information on the placement or mounting of the sensor in the bus driver’s cabin. It would be helpful to understand how the authors determined the optimal placement of the sensor and whether different locations were tested to evaluate the impact on data collection. Providing more details on the sensor placement process would enhance the understanding of potential influences on data accuracy and reliability.

- Dealing with sensor damage and changes in instrument placement is crucial for maintaining data quality and reliability. It would be valuable for the authors to elaborate on how they addressed or planned to address these issues. Discussing any measures taken to monitor and ensure the proper functioning of the sensors, as well as any protocols for addressing sensor damage or changes in placement, would provide insights into the steps taken to maintain data integrity.

- Considering factors such as vehicle movement, traffic patterns, and local sources of pollution is essential for accurate data processing and analysis. These factors can significantly impact the measured pollution levels and their spatial distribution. It would be beneficial for the authors to explicitly mention whether and how they considered these factors in their data processing and analysis. Neglecting these factors may lead to misinterpretation of the data and can affect the accuracy of the results.

**Documentation:**

See Limitations.

**Ethics:**

No.

**Limitations:**

- The lack of detailed information on the specific sensor selection process and calibration methods employed raises questions about the accuracy and reliability of the measurements obtained from the sensors.
- The paper does not address the potential impact of intermittent cellular connectivity on data completeness and accuracy. It is important to acknowledge and discuss any potential data loss or handling methods related to intermittent connectivity during bus travel.
- The paper does not explicitly acknowledge or discuss the potential biases introduced by focusing on specific bus routes/areas for data collection. This may limit the generalizability of the model, and it would be beneficial for the authors to address and discuss any biases or limitations associated with the selected bus routes/areas.

**Opportunities For Improvement:**

See Limitations

**Relation To Prior Work:**

Yes.

**Summary And Contributions:**

The paper addresses the issue of air pollution in developing countries, specifically India (Delhi), and the challenges of collecting fine-grained data due to the high cost of static sensors. To overcome this limitation, the authors propose a mobile sensor network using affordable PM2.5 sensors mounted on public buses in the Delhi-NCR region. They introduce a novel dataset containing PM2.5 and PM10 measurements, which is publicly available and valuable for both machine learning researchers and environmentalists.
In their study, the authors conducted statistical analysis and benchmarking to showcase the uniqueness and potential insights of their dataset. They compared it with existing pollution datasets (Canada dataset (Hamilton in Ontario, Canada Adams and Corr) and USA AQI dataset) and demonstrated substantial differences, underscoring the dataset's complexity and richness. The benchmarking exercise evaluated various interpolation, feature imputation, and forecasting methods (Mean Predictor, Inverse Distance Weighting (IDW), Random Forest (RF), XGBoost (XGB), ARIMA, N-BEATS, Non-Stationary Gaussian Process (NSGP) and Graphsage), highlighting the superior performance of the proposed dataset compared to others available.

---

> ### Author Response · Authors · 2023-08-25
> **Part 1/2 for reviewer npPd**
>
> We thank the reviewer for critical review of our work and providing us an opportunity for further improvements through constructive feedback. We also appreciate for recognizing our work for *the provision of publicly accessible PM2.5 and PM10 dataset for air pollution analysis and the provision of benchmarking exercise, evaluating state-of-the-art methods for interpolation, imputation, and forecasting on this dataset*.
>
> We have updated the main manuscript and the appendix to address these comments. The changes made are highlighted in blue colour. A comprehensive response to your feedback is presented below.
>
> **(L1) sensor selection process and calibration methods, accuracy and reliability of the measurements obtained from the sensors**
>
> In section 3.2, we compare the data collected with reference sensors, showing the accuracy and reliability in detail. In the data quality analysis in *Fig 2-3 (P4-5)* and through Anova statistical test in *Appendix C*, we show that our low-cost sensors are performing similar to high grade Dustrak and static sensors. We understand and appreciate the reviewer’s concern that these sensors may need early calibrations as compared to reference sensors. But the colocation experiments presented in *Fig 2* were performed in July 2021 after deployment and data collection was over, and the sensors were brought to the lab. We have analyzed them to be working sound after ~6 months in the polluted field. So, in our understanding, the presented dataset has minimal (if not none) impact, by the variance or calibration concerns.
>
> Also, these sensors are sensing the air the actual commuters breathe. Their readings, if any different due to the placement, should be more realistic and accurate than sensors placed at heights from the ground and distant from traffic intersections.
>
> **(L2) intermittent cellular connectivity, data completeness and accuracy**
>
> As discussed in *Section 3.1 (P4:L137-138)*, we use an SD card to store the data when cellular signal is unavailable and transmit the pending data as connectivity restores. So, there is a minimal chance of data loss (only in cases of SD card failure).
>
> **(L3) potential biases due to specific bus routes/areas**
>
> The bus routes selected by us covered the different types of areas, including green cover, residential, commercial etc, termed as static factors. Hence, we have tried our best to reduce the bias towards any particular factor.
>
> We acknowledge that the data is though limited to the region from routes of the bus as they go through arterial roads, without deeper penetration in residential areas. We are yet moving from high limitation of 3-4 spatial sensors points in the same vicinity to million+ spatial observation points with urban arterial route level limitations. We have discussed the limitation in *Section 5 Limitations and Future Work (P10:L351-358)*, and also added detailed bus-routes related analysis in *Appendix F*.

---

> ### Author Response · Authors · 2023-08-25
> **Part 2/2 for reviewer npPd**
>
> **[Additional Feedback]**
>
> **(1) placement or mounting of sensor in the bus driver’s cabin**
>
> The details of sensor are already discussed in a published work as below -
>
> Low Cost Platform Design for Pollution Measurement in Delhi-NCR using Vehicle-Mounted Sensors. Tanishka Goyal, Ankita Singh, Smriti Chhaya, Aditi Vikas, Poorva Garg, Ritika Malik, Rijurekha Sen. Mobicom '18
>
> We provide a small discussion on the sensor device for completeness in *Section 3.1* and have now properly marked the reference for the same *(P3:L121-122)*.
>
> After a deliberate discussion with traffic authorities, as pointed in *Section 3.1 (P3:L123-124 and P4:L140-141)*, a reasonable place to mount the device is driver’s cabin in order to *[a]* power connectivity *[b]* avoid theft and *[c]* ensure enough ambient air to measure PM.
> The sensor device placement pictures are shown in `https://www.cse.iitd.ac.in/pollutiondata`.
>
> **(2) Dealing with sensor damage, maintain data quality and reliability, monitor/ensure proper functioning of sensors, protocols for sensor damage, maintain data integrity**
>
> We have used the anomaly detection approach as discussed in *Appendix G* to automatically detect, flag and fix any sensor related issues. We have been using the below approaches during dataset collection to effectively keep validating the data being collected.
> * inter-sensor PM values variation at traffic intersections
> * samples recorded per minute
> * number of minutes each device is active in an hour
> * number of active hours in a day
> * samples recorded per region and intra-sensor PM values variation
>
> Hence, we have taken good care to validate the functioning of sensors during the data collection step is multiple ways and are confident for the collected data integrity.
>
> Regarding the inter-sensor variation, the buses with the sensors installed followed some common routes or crossed at fixed intersections, any high difference in the PM readings of two sensors is an indicator of (at least) one of them is malfunctioning. Further, the buses passed through the vicinity of standard high-cost sensors installed by the pollution board, the analysis over which is shown in *Fig 3(a)*, and the collected data is coming from functioning sensors. We even did a post deployment analysis by bringing the sensors to the lab and placing with a reference and calibrated DustTrak sensor, which is presented in *Fig 2*.
>
> **(3) Local factors (vehicle movement, traffic patterns, and pollution source) impact accurate data processing and analysis**
>
> The objective of the data collection is to identify the PM levels in the air which the commuters breathe. So, local sources of pollution and traffic induced pollution is inherent in the collected dataset.
>
> In *Appendix E*, we have added the PM2.5 analysis with meteorological factors and observed positive correlation with humidity and negative correlation with temperature/wind-speed. We also discuss high PM2.5 situations not explained by these factors and are due to external factors like stubble burning. Using the external data sources as mentioned in *Appendix E (L769-777)*, can be further used to infer deeper insights from the pollution dataset.
>
> *We thank you again for your time to understand our work and provide constructive insights. We trust that these revisions will satisfactorily address the concerns raised and elevate the overall quality of our work to an acceptable standard.*

---

### Official Review · Reviewer_5SWu · 2023-07-30
**An interesting contribution but too many ideas have been put into a single paper**

**Rating:** 4
**Confidence:** 5
**Correctness:** The data set is constructed in a soun…

**Strengths:**

1. The released dataset is interesting as it is a new mobile air quality data set from a region susceptible to air pollution episodes.
2. The work on the mobile air quality benchmark is a good starting point for a serious air quality model benchmark.


**Additional Feedback:**

I find this work to be a valuable dataset for the community, however releasing a data set in a way that makes it easily useful for other researches is a lot of work. I highly recommend continuing this work to improve the paper, I am not sure if you can manage to get this corrected for neurips, but do not give up - there are multiple venues that are well received where you can submit data sets. Please prepare an informative version of the data set paper, with exploratory data analysis, so that the community can best benefit from your work.

**Clarity:**

I find this paper to be a mixture of three ideas that deserve separate papers:

1. a sensor design with reference to the gold standard and a detailed analysis on how well the sensor follows the reference, which in itself is an important topic

2. a data set, for which it is not of that strong importance whether the sensor is very accurate or not, but rather - the characteristics of the data need to be presented, related to external characteristics such as weather, congestion, providing exploratory and explanatory analysis of what is happening, a dive into autocorrelation phenomena, and especially - explaining what is hard and easy to model in the data set (ex. an easy thing is when something is caught by very simple models, like repeating last value of MA(n), where n can be estimated from autocorrelation.

3. a benchmark, where a wide variety of data sets should be used to evaluate how a set models performs, with arguments for selecting data sets.

It is hard to read a paper which tries to make these three ideas fit a conference page limit. I would recommend to revise this paper thoroughly into either a data set paper, and develop that paper in detail. I have read papers that were trying to do such things alltogether, but they were journal papers well beyond a conference paper page limit.

**Documentation:**

No maintenance plans were included in the description.

**Ethics:**

I see not problems with ethical considerations, but elements related to what terms have been used to mitigate errors caused by measuring air pollution should be noted.

**Limitations:**

The limitations should be discussed, especially in terms of where the sensors made mistakes that can be detected by comparing to other data sources.

**Opportunities For Improvement:**

As this paper seems to mix the idea of a data set and a benchmark paper I will discuss the opportunities for improvements as a data set paper. In general the main area that can be improved is data set presentation and explanation. Please inform the community in detail about what the data set contains, in short: please provide an extensive data exploration alongside the data set - that is much more informative than doing a benchmark of algorithms on three data sets with inconclusive results.

1. Discuss air pollution episodes. Inform the readers about the % of non-pollution data in the data set and inform them about the air pollution episodes that are present in the data set. How many? When did they happen? Can you provide a cause for them? Are there any external data that can be useful when trying to model them, such as weather or traffic data? If so, are there perhaps any external data sources you can recommend for augmentation when your data set is used?

2. Discuss state transitions between air quality situations. What happens around the moments when the air quality gets worse, and when it gets better? How long does a transition last before a new pollution plateau is achieved. These are the places most complicated to model, it would be good to know them.

3. Discuss autocorrelation in various scenarios, obviously there will be autocorrelation for long periods of clean air, but what about pollution episodes? How often will a moving average model (and of what window size?) be correct? Plot autocorrelation lags. This tells people what to expect when putting a timeseries model on the data. Is there seasonality in the data set? Can you separate seasonality and trends and discuss them?

4. Discuss spatial lag and autocorrelation, for how long does a nearby location's reading predict a current location's air quality? Is that related to urban characteristics of the area? If so, inform about them. Discuss routes in more detail, perhaps separate them. And compare the histograms of their data sources?

5. Prepare a clean version of the raw data sets, with averaging per some window size to account for individual measurement errors, verify that maximum values are in some sensible range, ex. 1750 ug/m3 is quite a high measurement, there must be some reference stations in new delhi than can provide a sensible maximum value guidelines for air quality in the city. Remove windows or cap values with clear mearument errors, and as we all know - these will happen more in the high pollution zones. Double check GPS values, perhaps snap gps values to route if gps errors move points far away from route.

6. Either do not include a benchmark, leave that for another paper, this data exploration will already take a lot of space, or include a benchmark but submit a much larger paper to a journal, see STOTEN, Scientific Data, Earth System Science Data, Geoscience Data Journal, Atmosphere, IEEE Transactions on Instrumentation and Measurement, many machine learning journals also take data+benchmark papers. It's easier to fit such an analysis without a page limit.

7. When you do benchmarking please consider reporting mean average and mean square errors per air quality interval, so that we can have some kind of orientation in how a model performs at a given air quality situation. Preferably also inform about how well the model performs at sharp air quality changes, i.e. select an air quality index, or just jenks intervals and report mae/mse on the border between them.

8. For the clean version of the data set, propose a balanced stratification, so that we can get reproducible results. Provide a benchmark website with reproduction instructions, ranking tables for models and a procedure for reporting new model's results (ex. a pull request with a github PR edit).

9. Consider uploading the data set in both clean and raw versions to huggingface, so that it can be easily loaded by people with less knowledge about time series or geo series, but with knowledge about modelling.

**Relation To Prior Work:**

There are remarkably few papers on data sets for mobile air quality collection, and the one's that are mentioned are well selected. What is missing in the benchmark part is that there's been a lot of literature on doing benchmarks, benchmarks with time series and geoseries too, including discussion on continous benchmarking, balancing data splits, releasing clean data, new model submission procedures, etc. One can learn a lot from how people in NLP or CV have been developing their benchmarks in recent years. I expect this to be a data paper, but if the authors decide to separate data and benchmark and publish the benchmark somewhere, including a background dicussion on what makes a good benchmark and how those elements of a good benchmark are fulfillled in the benchmark construction.

**Summary And Contributions:**

The authors propose a new data set for air quality prediction which includes 3 months of data collected from an optical sensor mounted on public buses. The paper also includes a comparison of several deep learning and classical prediction methods for three air quality prediction tasks.

---

> ### Author Response · Authors · 2023-08-25
> **Part 1/3**
>
> We thank the reviewer for critical review of our work and providing us an opportunity for further improvements through constructive feedback. We appreciate for recognizing our work on the mobile air quality benchmark *to be a good starting point for a serious air quality model benchmark, with the released dataset as a new mobile air quality data set from a region susceptible to air pollution episodes.*
>
> We also appreciate your suggestion for pushing our work as a dataset paper. We agree that the main focus of our work is to present a novel dataset. However, as our dataset is unique in terms of the distribution and novel challenges it brings for modeling as compared to other datasets available in the public domain, we hence performed benchmarks to show that our dataset is harder to model compared to public datasets.
>
> To improve the data exploration part, we have now added the below -
>
> * Dataset analysis for correlation in PM2.5 *(Appendix D)*
> * Analysis with meteorological factors *(Appendix E)*
> * Different bus-routes related analysis *(Appendix F)*
> * Anomaly detection *(Appendix G)*.
>
> We have updated the main manuscript and the appendix to address the feedback. The changes made are highlighted in blue colour. A comprehensive response to your feedback is presented below.
>
> **(Q1) Discussion on non-pollution data, air pollution episodes, when and why they happen, external data useful for modeling**
>
> Along the 3 pollution components PM1, PM2.5 and PM10, our dataset contains 7 additional parameters, as latitude, longitude, time, deviceId, pressure, temperature, humidity. We have mentioned this in Section 3.1 *(P4:L144-146)* and in Table 1 *(P5)*.
>
> There are several pollution episodes in this dataset (as visible in Fig 15 in Appendix E), the prominent one being November 5-10, 2020. As per [Weather](https://weather.com/en-IN/india/pollution/news/2020-11-10-delhi-pollution-capital-records-emergency-levels-of-air-pollution), there was high moisture, calm winds and stubble burning around this time in North India. Also, as per [HindustanTimes](https://www.hindustantimes.com/cities/pollution-in-november-2020/story-34vLB8FK1dAgcFazV69jEJ.html), Delhi had six consecutive severe days from November 5-10, 2020, the longest severe spell seen in the city since 2016, due to a prolonged and intensive stubble burning season with unfavourable meteorological conditions and other factors. We have discussed this in Appendix E along the meteorological factor analysis *(L757-762)*.
>
> Several external data can be useful while modeling our dataset. Some mentions are
> * Meteorological data from ERA5
> * Fire count from NASA VIIRS
> * Other Pollution Data from OpenAQI/CPCB
> * Photochemical modelling from www.camx.com
> * Planetary boundary layer height from ISRO
> * Traffic Data
>
> We have added the above (with proper references) in Appendix E *(L769-777)*.
>
> **(Q2) Discuss state transitions between air quality situations, worse and better air quality, places most complicated to model**
>
> Appendix E discusses the positive and negative correlation, intuition for the correlation, and counter-situations for the correlation observed between PM2.5 and meteorological parameters in Fig 15. Temperature expands the air reducing PM per unit volume, humidity/moisture intensifies it, and wind blows it away. However, at a given time, which meteorological factor among these may have dominated needs more complex ML modeling *(L746-752)*.
>
> Also, as already discussed in Q1 above, the episodes of worse air quality are attributed to external factors like intensive stubble burning season and the unfavourable meteorological conditions. In contrast, favourable winds (in speed and direction) and other improved meteorological factors would take the high pollution component to places away from the city and pave way for better air quality.
>
> In Appendix H, we have also analyzed the dataset to find spatial locations presenting high modeling errors, which is shown in Fig 24.
>
> **(Q3) Discuss autocorrelation in various scenarios, plot autocorrelation lags, seasonality**
>
> Over a short duration, there is a high autocorrelation for 1-hour lags, thereafter the auto-correlation drops. Over longer term, we have observed two kinds of locations, one is the bus depot where the buses stop between transits and overnight, the PM pattern here correlates to 24-hour windows. Other are grid locations throughout the bus routes, where data also shows 8 or 12 hour window auto-correlation, in lesser capacity than the 24 hour window.  The detailed analysis is presented in Appendix D.
>
> Also, as this dataset is collected in winter season, it contains the influence of related meteorological and seasonal factors as discussed in Appendix E and expressed in Q2 above.

---

> ### Author Response · Authors · 2023-08-25
> **Part 2/3**
>
> **(Q4) Discuss spatial lag and autocorrelation, nearby location's impact on air quality, discuss routes in detail and separately**
>
> In *Appendix D*, we analyzed the autocorrelation and covariance over the spatial locations. We observe high correlation with 1 hour and 24-hour lags for all locations, with additional 8 or 12 hours lags for some locations. We also observe covariance among different spatial locations.
>
> Alongside, the nearby location influence on the pollution traits is evident from the metrics of Inverse Distance Weighting (IDW) algorithm present in the benchmarking exercise in *section 4.4*. This algorithm predicts the PM of a location taking the PM levels of nearby locations, factored by distance. The simple Mean Predictor does not take distance into account. So, the impact of an adjacent location is maximum for IDW w.r.t Mean Predictor. In *Fig 5 (P8)*, for Delhi data, we can observe that IDW performs significantly better than the Mean Predictor, pointing that air quality is highly influenced by nearby locations. Thankful to your insights, we have added this as *Observation 5 (P9:L328-332) in section 4.4*.
>
> We have also performed route level analysis and presented the work in *Appendix F*. We have shown the spatial PM2.5 levels *(Fig 16-17)*, temporal PM2.5 levels *(Fig 18)*, and observed similar autocorrelation *(Fig 19)* and covariance *(Fig 20)* traits as with the combined analysis presented in *Appendix D*. In contrast to the equally positive and negative correlation for the spatial grid autocorrelation in *Fig 19*, we observed less negative autocorrelation for the bus-routes, which seems due to the same paths being traversed at same time each day by the buses.
>
> **(Q5) Clean dataset version, validate GPS values**
>
> The GPS readings are usually accurate enough to visually show the buses enroute over the map. To handle for the small GPS inaccuracies, the off-shoot readings are corrected (interpolated) or discarded. We do not clip the PM values taking reference from nearby stations, as these sensors are working at ground level where the commuters actually breathe. A temporary extremely high pollution at a traffic signal due to high-polluting vehicles in the vicinity, is a possibility. Hence, we retain the peculiar characteristics of the collected data. In the cleaned data, we have a max PM2.5 of 1448, with 0.11% samples greater than 800 PM2.5. When we apply gridding to the data, the grouping of PM values over a window softens the singular high measurements taking their overall effect to minimal, while retaining instances of actual high PM2.5 spikes. In the gridded data, we have only 6 instances above 800 PM2.5 (1113, 931, 816, 822, 880, 872), which could be further clipped or omitted as per the modeling requirements.
>
> As per your good advice, we have uploaded all the dataset variants (raw, cleaned, and gridded) at `https://huggingface.co/datasets/sachin-iitd/DelhiPollDataset`, along with the meteorological parameters.
>
> **(Q6) Omit benchmark to leave that for another paper, the data exploration will already take a lot of space**
>
> The benchmarking is utilized to support the presented dataset analysis. Using minimal space for other datasets, we tried to highlight the attributes of our dataset in contrast with others. *The benchmarking is also appreciated by other reviewers.* We believe that our dataset and presented analysis around core ML problems of interpolation and forecasting is highly aligned with NeurIPS despite being page limited.
>
> The objective of the track is to accept novel datasets, and the benchmarking against the simple Canada and USA datasets exhibit the complexity, uniqueness, and novelty of our dataset, alongside the data exploration (original and new) performed.
>
> **(Q7) With benchmarking, report mean average and mean square errors per air quality interval, model performance at sharp air quality changes, report mae/mse on the border between them.**
>
> Our dataset contains raw PM2.5 values which are not splitted like AQI bands and are used directly for the benchmarking. The MAE/RMSE plot for different PM levels is added in *Fig 6 (P8) of section 4.4*. We have observed that most errors are present at high PM levels. There seems a linear increase in error for low PM levels (which may also lie at the AQI boundaries.) This error plot also shows the Delhi dataset is harder to model as compared to the existing Canada / USA datasets, due to high PM2.5 levels.

---

> ### Author Response · Authors · 2023-08-25
> **Part 3/3**
>
> **(Q8) Reproducible results, benchmark website, procedure for reporting new model's results**
>
> **(Q9) Upload dataset to huggingface**
>
> The gridded version of the dataset (for 1km x 1km x 1hr) contains one sample per cell. We have undertaken experiments for the test data selection using multiple approaches, like
> * take all temporal samples for some spatial locations (leaving none for train set)
> * take limited samples from all spatial locations
> * random selection/split
>
> We found similar benchmarking performance in the above and decided to take random split as our default choice for all experiments. We are hopeful that the benchmarking would show similar results for the balanced stratification as you would suggest.
>
> As per your good suggestion, we have uploaded the raw, clean, and gridded version of the dataset to `https://huggingface.co/datasets/sachin-iitd/DelhiPollDataset`, along with test/train splits for reproduction. We also have a dataset website `https://www.cse.iitd.ac.in/pollutiondata`. which we will extend with benchmarking analysis and provide procedures for reporting new model's results.
>
> **[Limitations] Detect sensors' mistakes**
>
> We have discussed the limitations (as also suggested by other reviewers) under *section 5 - Limitations and Future Work (P10:L351-358)* and other appropriate places like *section 3.1 (P3:L132-134)*.
>
> For the sanity of the collected data, the buses with the sensors installed followed some common routes or crossed at fixed intersections, any high difference in the PM readings of two sensors is an indicator of (at least) one of them is malfunctioning, which was taken care in data collection/cleaning. Also, the buses passed through the vicinity of standard high-cost sensors installed by the pollution board, the analysis over which is shown in section 3.2 Fig 3(a) *(P5)*. The post deployment colocation experiments presented in section 3.1 Fig 2 *(P4)* and Fig 9 in Appendix C also shows the soundness of sensors after the data collection is over.
>
> We have further added our anomaly detection approach in *Appendix G* used to automatically detect and flag any related issues. We check for the below during dataset collection to effectively keep validating the data being collected.
>
> * inter-sensor PM values variation (discussed above)
> * samples recorded per minute
> * number of minutes each device is active in an hour
> * number of active hours in a day
> * samples recorded per region
> * intra-sensor PM values variation
>
> **[Clarity] Mixture of three ideas that deserve separate papers**
>
> We aim to present our work as a dataset paper.
>
> The design part is already published and properly cited in Section 3.1. The current paper just describes the basics of the device (in less than a page) for completeness purpose.
>
> As expressed in previous points, benchmarking against existing Canada and USA datasets is necessary to show the complexity of our dataset and highlight its novelty and uniqueness.
>
> As also expected by other reviewers, we prefer to showcase that our data is coming from reliable sensors and is accurate enough to validate its utility for deeper insights generation beyond ML modeling. To improve the data exploration part, we have now added dataset analysis for correlation in PM2.5 *(Appendix D)*, analysis with meteorological factors *(Appendix E)* and different bus-routes related analysis *(Appendix F)*, explaining some insights and observations.
>
> **[Relation To Prior Work]**
>
> We appreciate your suggestions on improving the benchmarking part. As expressed in previous points, we believe that our work suits the dataset track, and the contrast benchmarking is in the same alignment to complement the data exploration and explanation for the given dataset.
>
> *We thank you again for your time to understand our work and provide constructive insights. We trust that these revisions will satisfactorily address the concerns raised and elevate the overall quality of our work to an acceptable standard.*

---

> > ### Author Response · Authors · 2023-08-29
> > **Eagerly waiting for your feedback on our revisions**
> >
> > Dear Reviewer,
> >
> > We thank you for taking the time to provide constructive comments, which have significantly improved the quality of the manuscript. Since we are just a few hours away from the closure of the author-reviewer discussion period, we are keen to know if there are any outstanding concerns that remain to be addressed. We have conducted a rigorous rebuttal and hope the reviewer will be convinced of the merits of our work.
> >
> > We are also happy to inform you that one of the reviewers has now increased the rating to Strong Accept following our revision.
> >
> > regards,
> >
> > Authors

---

### Author Response · Authors · 2023-08-25
**A note of thanks**

We express our gratitude to the reviewers for their perceptive insights and constructive suggestions. We also appreciate the reviewers for recognizing our work -

**(5SWu)** on the mobile air quality benchmark to be a good starting point for a serious air quality model benchmark, with the released dataset as a new mobile air quality data set from a region susceptible to air pollution episodes.

**(npPd)** for the provision of publicly accessible PM2.5 and PM10 dataset for air pollution analysis and the provision of benchmarking exercise, evaluating state-of-the-art methods for interpolation, feature imputation, and forecasting on this dataset.

**(Q5NU)** for being a well-motivated paper to addresses a pressing real-world challenge, with well-stated contributions and (good) data collection frequency, having good implications, with the method being used to recommend commute/travel routes.

**(8Gda)** for being different from previous dynamic sensing networks in highly polluted cities like Delhi, which is valuable for future work studying pollution, with ML interpolation and forecasting benchmarks showing that traditional ML methods are sufficient to learn the patterns in the data.

A comprehensive point-by-point response to each reviewer's comments is presented separately below. We have updated the main manuscript and the appendix to address these comments. The changes made are highlighted in blue colour. The major additional experiments and changes, to effectively address the raised concerns, are listed below.

1. Correlation Analysis in PM2.5 *(Appendix D)*
2. Analysis with meteorological factors *(Appendix E)*
3. Different bus-routes related analysis *(Appendix F)*
4. Anomaly detection *(Appendix G)*
5. Reordering of sections 4.1, 4.2 and 4.3 to improve the structure and readability, based on reviewer's (Q5NU) suggestion
6. Discussed Limitations on Page 10 under *Section 5 – Limitations and Future Work*
7. Upload different variants of PM dataset (with Meteorological data) to `https://huggingface.co/datasets/sachin-iitd/DelhiPollDataset`

We trust that these revisions will satisfactorily address the concerns raised by the reviewers and elevate the overall quality of our work to an acceptable standard.

---

### Decision · Program_Chairs · 2023-09-22

**Decision:**

Accept (Poster)

**Comment:**

We seldom come across datasets where the data creators capture the data themselves rather than extract them from some system designed by a different team. This is an example of what Kiri Wagstaff referred to as "machine learning that matters".

The reviewers generally pointed out that the paper is well-motivated, addressing a pressing real-world challenge, the execution is well-done, and the work is impactful. Furthermore, the mobile air quality benchmark can be used as a starting point for future benchmarks

There was one reviewer who did not have a positive view of the work, but they did not revise their review after the author rebuttal, and therefore their review was weighted less.

Overall, I recommend acceptance.